
# Bethe $M$-layer construction for the percolation problem

Maria Chiara Angelini[1,2], Saverio Palazzi[1*],
Tommaso Rizzo[3,1] and Marco Tarzia[4,5]

**1** Dipartimento di Fisica, Sapienza Università di Roma,
Piazzale A. Moro 2, I-00185, Rome, Italy
**2** INFN-Sezione di Roma 1, Piazzale A. Moro 2, 00185, Rome, Italy
**3** ISC-CNR, UOS Rome, Sapienza Università di Roma,
Piazzale A. Moro 2, I-00185, Rome, Italy
**4** LPTMC, CNRS-UMR 7600, Sorbonne Université, 4 Pl. Jussieu, F-75005 Paris, France
**5** Institut Universitaire de France, 1 rue Descartes, 75231 Paris Cedex 05, France

⋆ saverio.palazzi@uniroma1.it

## Abstract

One way to perform field theory computations for the bond percolation problem is through the Kasteleyn and Fortuin mapping to the $n + 1$ states Potts model in the limit of $n \to 0$. In this paper, we show that it is possible to recover the $\epsilon$-expansion for critical exponents in finite dimension directly using the $M$-layer expansion, without the need to perform any analytical continuation. Moreover, we also show explicitly that the critical exponents for site and bond percolation are the same. This computation provides a reference for applications of the $M$-layer method to systems where the underlying field theory is unknown or disputed.



# 1  Introduction

The percolation problem provides one of the simplest examples of a second-order phase transition, in both the versions of site or bond percolation. Despite the simplicity of the model, it is at the basis of different problems in many different fields, from condensed matter to telecommunication engineering, from graph theory to epidemic spreading [1,2]. In the standard site (bond) percolation problem, each site (bond) is present independently of the neighbors with probability $p$. Above a certain threshold $p_c$, a giant cluster of nearest-neighbor sites is present in the thermodynamic limit while below this threshold neighboring sites are grouped into many small clusters of non-extensive size. The value $p_c$ corresponds to the transition point and one can associate standard critical exponents that describe how critical observables behave near $p_c$. Despite the deep similarities with respect to critical behavior, the main difference between percolation and other phase transition models is the absence of an associated Hamiltonian and a corresponding partition function.

The renormalization group (RG) is the main tool to study second order phase transitions. It can be applied in two ways: the first one is by performing explicitly an RG transformation on a given two- or three-dimensional lattice while the second relies on field theory. The first method typically requires uncontrolled approximations (needed to close the RG equations and find a fixed point) while the second is more powerful as it allows one to systematically obtain the critical exponents in dimension $D$ in powers of $\epsilon = D_U - D$ where $D_U$ is the upper critical dimension. The first method can be applied to percolation as it is [3,4] but one could think that the lack of a Hamiltonian would make the application of the second impossible. However, in a seminal paper, Kasteleyn and Fortuin showed that the bond percolation problem is exactly related to the $n \to 0$ limit of an $n$-component ($n + 1$ states) Potts model [5]. It was then recognized [6] that this mapping allows the application of field-theoretical techniques and today the exponents are known up to the 5th order in an $\epsilon$-expansion around the upper critical dimension [7–11].

In this paper, we reproduce the same expansion up to one-loop order by means of the $M$-layer construction. This construction has been introduced in Ref. [12], and then applied to a variety of models [13–18]. The useful property of the $M$-layer construction is that one can also study the critical behavior, in finite dimensions, of problems which are not defined by a Hamiltonian, such as the percolation. One has to introduce $M - 1$ independent lattices, in addition to the original one; the $M$ layers will then be coupled together through a random rewiring of the bonds. The $M \to \infty$ limit gives the Bethe lattice solution [19] of the model, while if $M = 1$ one obtains the original model. An expansion in $1/M$ can be properly set up, that is in practice an expansion in the number of the topological loops considered. The $M$-layer construction can be applied to any model that can be defined on a locally tree-like graph, including percolation. This is interesting, because, with this approach, there is no need to invoke the $n \to 0$ analytic continuation discovered by Kasteleyn and Fortuin. Furthermore, with this method, we can also analytically verify that the critical exponents of site percolation are equal to those of bond percolation.

The additional value of this paper is methodological: we show for the first time that from the $1/M$ expansion on the $M$-layer lattice one can obtain the $\epsilon$-expansion, through the suitable introduction of a dimensionless beta function in analogy with what is usually done in standard field theory [20, 21]. This is a fundamental step that will help in applying in the future the same techniques to more complicated systems, for which a finite-dimensional solution is still not known, such as the Edward-Anderson spin-glass model [17] or Anderson localization [18].

The paper is organized as follows: In Section 2 we present the model and the main results, in particular we sketch the derivation of the $\epsilon$-expansion for the critical exponents from the $1/M$ expansion of two- and three-point correlation functions. In Section 3 we introduce the problem on the Bethe lattice with a novel derivation of the cluster distribution function. In Section 4 we recall the general properties of the $1/M$ expansion and the operative rules to compute it. In Section 5 we present the computation of the observables in the $M$-layer framework for both site and bond percolation. In Section 6 we apply one of the standard methods to compute critical exponents in $\epsilon$-expansion. Finally, in Section 7, we give our conclusions.

## 2 Models and main results

In this Section we list the results of the application of the $M$-layer construction to both the bond and site percolation problems on a hyper-cubic lattice in $D$ dimensions. We briefly describe the steps needed to reach the final results which will be summarized next.

In the standard site (respectively bond) percolation problem, each site (respectively bond) is present, or "active", independently of the neighbors with probability $p$. In the site percolation problem one then defines a cluster as a subset of nearest-neighbor active sites, while in bond percolation a cluster is defined as a subset of sites connected by nearest-neighbor active bonds. At $p_c$ a giant cluster appears, that contains a finite fraction of all the sites $N$. Our analysis will mainly apply to the non-percolating phase $p < p_c$ and from now on we refer to this case. The critical behavior in the non-percolating phase is characterized by considering the average number $n(s, p)$ of clusters of size $s$ in a system of size $N$. This distribution is cut off at a typical size $s^*$ that diverges at the critical point. We also consider the $q$-point function $C_q(x_1, \ldots, x_q)$ that gives the probability that the sites at $x_1, \ldots, x_q$ belong to the same cluster. According to scaling arguments [1, 22], we expect that the two-point function obeys the following scaling form for large $|x_1 - x_2|$ and for $p$ close to $p_c$:

$$C_2(x_1, x_2) = \frac{1}{|x_1 - x_2|^{D-2+\eta}} f_{C_2}\left(\frac{|x_1 - x_2|}{\xi}\right), \tag{1}$$

where $f_{C_2}$ is a proper scaling function, $\eta$ is the anomalous dimension and $\xi$ is the correlation length that diverges at the critical point as:

$$\xi \sim \frac{1}{|p - p_c|^\nu}. \tag{2}$$

The typical size $s^*$ scales with the correlation length as

$$s^* \sim \xi^{D_f}, \tag{3}$$

where $D_f$ stands for the fractal dimension of the clusters. The distribution of the cluster sizes also obeys a scaling law [1, 22]:

$$n(s, p) = s^{-\tau} f_n\left(|p - p_c|s^\sigma\right), \tag{4}$$

where $f_n(x)$ is another scaling function. We also consider the space integrals of the $C_q(x_1,\ldots,x_q)$, called susceptibilities,

$$\chi_q \equiv \sum_{x_2,\ldots,x_q} C_q(x_1,\ldots,x_q), \tag{5}$$

that are independent of $x_1$ in a homogeneous system (they only depend on the differences between the points). They are related to the moments of the $n(s,p)$ through:

$$\chi_q = \sum_{s=0}^{\infty} s^q\, n(s,p). \tag{6}$$

The scalings of the typical size $s^*$ and the correlation length $\xi$ give

$$\sigma = \frac{1}{\nu D_f}, \tag{7}$$

while, given the relation

$$\tau = 1 + \frac{D}{D_f}, \tag{8}$$

comparing Eqs. (5) and (6) and using the scaling of $n(s,p)$ one can easily find that the susceptibilities diverge as

$$\chi_q \sim \xi^{-D+D_f q}, \tag{9}$$

from which it follows that the following quantity goes to a constant at the critical point:

$$\lambda \propto \xi^{-D} \frac{\chi_3^2}{\chi_2^3}. \tag{10}$$

On the $M$-layer lattice $\chi_2$ and $\chi_3$ are given by the Bethe lattice solution in the limit $M \to \infty$ and we computed the first $1/M$ correction, for both site and bond percolation. Once the two-point observable is computed, with the $M$-layer construction, the upper critical dimension, $D_U$, can be deduced from the Ginzburg criterion and for the percolation problem it turns out to be $D_U = 6$. At this point of the computation a standard procedure to compute critical exponents is applied [20]. In particular we write $\lambda$ as:

$$\lambda = u - \frac{7}{4} \frac{u^2}{(4\pi)^{\frac{D}{2}}} \Gamma\left(3 - \frac{D}{2}\right) + \mathcal{O}(u^3), \tag{11}$$

where the constant $u$ is defined as $u \equiv g\, m^{D-6}$, where $m \equiv \xi^{-1}$ and $g$ is a $\mathcal{O}(1/M)$ constant that depends on the microscopic details of the model including whether we consider bond or site percolation. Note that the dimensionless constant $u$ diverges at the critical point for $D < 6$ because $m$ vanishes, while $\lambda$ remains finite at the critical point according to Eq. (9). Notice that, in order to understand this last statement from Eq. (11), one should consider the relation between $\lambda$ and $u$ to all orders in $u$, but in this perturbative framework we only compute the first correction, to $\mathcal{O}(u^2)$. We expect that:

$$\lambda \approx \lambda_c + c_1 \xi^\omega = \lambda_c + c_1 m^{-\omega}, \quad \text{for} \quad \xi \to \infty,\, m \to 0, \tag{12}$$

where $c_1$ is a model-dependent constant, while $\omega$ is a universal exponent that controls the corrections to scaling [20]. Now, following a standard field-theoretical procedure (see Ref. [20], Chap. 8), we define the function $b(\lambda)$, using the above relationships:

$$b(\lambda) \equiv m^2 \frac{\partial}{\partial m^2} \lambda \approx -\frac{\omega}{2} c_1 m^{-\omega} \approx -\frac{\omega}{2}(\lambda - \lambda_c), \tag{13}$$

meaning that at the critical point

$$b(\lambda_c) = 0, \ \omega = -2 \, b'(\lambda_c). \tag{14}$$

From (11), we obtain an expression of $b(\lambda)$ to second order in $\lambda$ from which the following scenario emerges: for $D \geq D_U = 6$ only the solution $\lambda = 0$ exists, meaning that $\lambda$ tends to zero at the critical point with $\omega = 6 - D$, while for $\epsilon \equiv 6 - D > 0$ a new solution $\lambda_c \neq 0$ appears:

$$\lambda_c = \frac{2(4\pi)^3}{7} \epsilon + O(\epsilon^2), \tag{15}$$

and $\lambda$ tends to $\lambda_c$ at the critical point, with $\omega = -\epsilon + O(\epsilon^2)$. Here the universality is realized: the non-trivial fixed point $\lambda_c$ doesn't depend anymore on the specific value of $g$ and thus it doesn't depend on the microscopic details of the system (including if we are dealing with bond or site percolation). Moreover we confirm that, due to universality, the values of the critical exponents do not depend on the value of $M$.

Following similar standard computations (see Ref. [20], Chap. 8), from the value of $\lambda_c$ and the scaling laws, we obtained the $\epsilon$-expansion for the critical exponents:

$$\nu = \frac{1}{2} + \frac{5}{84} \epsilon + \mathcal{O}(\epsilon^2), \tag{16}$$

$$\eta = -\frac{1}{21} \epsilon + \mathcal{O}(\epsilon^2). \tag{17}$$

Comparing Eq. (9) with the scaling law $\chi_2 \sim \xi^{2-\eta}$ we obtain

$$D_f = \frac{D + 2 - \eta}{2}, \tag{18}$$

all the other critical exponents can be obtained from $\eta$ and $\nu$ through the scaling laws given above.

We stress that the result is independent of the actual values of any non-universal constant, ensuring that the critical exponents are the same for bond and site percolation, as explained more extensively in Sec. 5. As it should, the results coincide with those obtained from the $\epsilon$-expansion for the $(n+1)$-state Potts models in the limit $n \to 0$, which coincides with bond percolation according to the Fortuin-Kasteleyn mapping. In appendix D we have also computed the expansion of $\chi_4$ in powers of $1/M$ checking that it diverges at the critical point with an exponent equal to that predicted by Eq. (9).

## 3 Percolation on the Bethe lattice

In this Section we show how to derive equations for the critical behavior of $g(s, p)$, defined as the probability that a randomly chosen site belongs to a cluster of size $s$, including $s = 0$ meaning that the site is not active. We discuss the case of site percolation on a Bethe lattice and how to derive the exact critical exponents in this case. Given the definition of $n(s, p)$, in Sec. 2, we have

$$g(0, p) = (1 - p), \ g(s, p) = s \, n(s, p), \quad \text{for} \quad s > 0. \tag{19}$$

Here and in the following we call "Bethe lattice" a random regular graph with fixed connectivity $c$. Notice that $g(s, p)$ as it should is normalized to 1 because the probability that a randomly chosen site belongs to a cluster is $\sum_s s \, n(s, p) = p$. We also define the associated "cavity" probability, $g_{cav}(s, p)$, as the probability that a randomly chosen site, for which one of its $c$ edges is removed, belongs to a cluster of size $s$. This definition is useful since on a Bethe

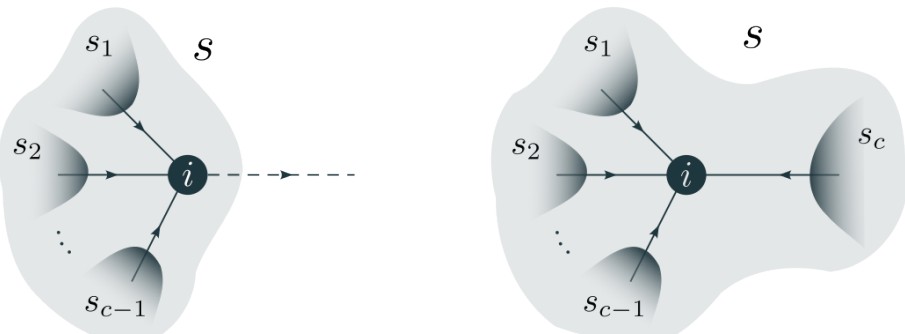

Figure 1: Graphic representation of Eqs. (20) and (21). Left: on a Bethe lattice with connectivity $c$ one of the edges of site $i$ is removed, represented with a dashed line. The $c-1$ remaining neighboring sites are connected by site $i$ only, thus the probabilities $g_{cav}(s_1, p), \ldots, g_{cav}(s_{c-1}, p)$ are factorized. The total resulting size is $s = 1 + s_1 + \cdots + s_{c-1}$. Right: in this case none of the edges of site $i$ is removed. Again the cavity probabilities are factorized, but in this case the product includes $g_{cav}(s_c, p)$ too.

lattice two sites are connected by a unique sequence of adjacent edges, so that, removing one edge, the two sites will be completely independent and the resulting probability to belong to a cluster factorizes [23]. Thus, given that each site is active with probability $p$, we can write a self-consistent equation for $g_{cav}(s, p)$ on the Bethe lattice with fixed connectivity $c$:

$$g_{cav}(s,p) = (1-p)\,\delta_{s,0} + p \sum_{s_1=0}^{\infty} \cdots \sum_{s_{c-1}=0}^{\infty} g_{cav}(s_1,p) \ldots g_{cav}(s_{c-1},p)\delta_{s,1+s_1+\cdots+s_{c-1}}, \qquad (20)$$

where the first term comes from the case in which the given site is not present, and the resulting size of the cluster is $s = 0$, while the second is the probability that the site is present, $p$, times the product of the factorized probabilities that the $c-1$ neighboring sites (one edge is removed, see Fig. 1) belong to clusters of sizes $s_1, s_2, \ldots, s_{c-1}$. In this second case the resulting size must be the sum of the sizes plus one, the given site. The probability $g(s, p)$ can then be expressed in terms of the cavity probability as:

$$g(s,p) = (1-p)\,\delta_{s,0} + p \sum_{s_1=0}^{\infty} \cdots \sum_{s_c=0}^{\infty} g_{cav}(s_1,p) \ldots g_{cav}(s_c,p)\delta_{s,1+s_1+\cdots+s_c}, \qquad (21)$$

the only difference being that the product is over $c$ terms $g_{cav}$, since all the $c$ edges of the given site are present. Next we define the generating function $\tilde{g}(t, p) \equiv \sum_{s=0}^{\infty} g(s, p)e^{-ts}$ and its cavity counterpart, $\tilde{g}_{cav}(t, p)$. Eq. (20) becomes:

$$\tilde{g}_{cav}(t,p) = (1-p) + p\left(\tilde{g}_{cav}(t,p)\right)^{c-1} e^{-t}. \qquad (22)$$

Deriving the above equation with respect to $t$ and setting $t = 0$ we obtain

$$\tilde{g}'_{cav}(0,p) = \frac{p}{p(c-1)-1}. \qquad (23)$$

The moments of $g(s, p)$ are related to the derivatives of $\tilde{g}(t, p)$ in $t = 0$, in particular, recalling the definition (6) we have:

$$\chi_2(p) = -\tilde{g}'(0,p) = \frac{p(p+1)}{1-p(c-1)}, \qquad (24)$$

that diverges, as expected, at the critical point $p_c = 1/(c-1)$. It is possible to obtain the previous divergent behavior considering the two-point correlation $C_2(x_1, x_2)$ defined in the previous Section. As we will see in Sec. 5, the correlation between two sites at distance $L$ on the Bethe lattice is $C_2(|x_1 - x_2| = L) = p\, p^L$. The associated susceptibility, $\chi_2$, turns out to be

$$\chi_2 = \sum_{x_2} C_2(|x_1 - x_2|) = p + p\sum_{L=1}^{\infty} c\,(c-1)^{L-1}\,p^L = \frac{p(p+1)}{1-p(c-1)}, \tag{25}$$

where the sum over $x_2$ is over all the sites of the Bethe lattice and $c\,(c-1)^{L-1}$ is the number of neighboring sites at distance $L \geq 1$. We notice that in the Bethe lattice the two-point correlation is always exponentially decaying, also at the critical point $p = p_c$. The reason for the divergence is the number of neighboring sites which is exponential in the distance $L$. This means that the correlation length $\xi_{BL}$, implicitly defined by

$$C_2(L) \propto p^L \equiv e^{-\frac{L}{\xi_{BL}}}, \tag{26}$$

is always finite and equal to $\left(-\log(p)\right)^{-1}$. With this definition of $\xi_{BL}$ the anomalous dimension, $\eta$, associated to the power law behavior of $C_2(L)$, and the exponent $\nu$ associated to the power law behavior of the correlation length, are not defined on the Bethe lattice.[1] One of the interesting features of the $M$-layer construction is that it allows to compute $\eta$ and $\nu$ also in the limit $M \to \infty$, as we will discuss later.

**Scaling of $n(s, p)$**  In order to compute the scaling of $n(s, p)$, we are interested in the functions $g(s, p)$ for $p$ close to the critical point and $s$ large, that corresponds to small values of $t$ in $\tilde{g}(t, p)$. We now define

$$\delta\tilde{g}(t, p) \equiv \tilde{g}(t, p) - 1 = \sum_{s=0}^{\infty} g(s, p)(e^{-s\,t} - 1), \tag{27}$$

and its cavity counterpart $\delta\tilde{g}_{cav}(t, p) \equiv \tilde{g}_{cav}(t, p) - 1$. Differentiating Eq. (22) with respect to $t$ we obtain, for small values of $t$ and $p$ close to $p_c$:

$$\delta\tilde{g}'_{cav}(t, p)\big(1 - p/p_c - (c-2)\delta\tilde{g}_{cav}(t, p)\big) = -p_c, \tag{28}$$

from which we have

$$\delta\tilde{g}_{cav}(t, p) = a\left(1 - (1 + t/t^*)^{1/2}\right), \tag{29}$$

where

$$\delta p \equiv p - p_c, \quad a \equiv -\delta p\,\frac{c-1}{c-2}, \quad t^* \equiv \delta p^2\,\frac{(c-1)^3}{2c-4}. \tag{30}$$

For small values of $t$ and $\delta p$ we also obtain

$$\delta\tilde{g}(t, p) = \frac{c}{c-1}\delta\tilde{g}_{cav}(t, p). \tag{31}$$

Replacing the sum with an integral (which is justified by the fact that small values of $t$ correspond to large values of $s$) we obtain, computing the inverse Laplace transform of Eq. (29) and using Eq. (31)

$$g(s, p) \sim \frac{1}{s^{3/2}}e^{-s\,t^*} \to n(s, p) \sim \frac{1}{s^{5/2}}e^{-s\,t^*}, \tag{32}$$

---

[1] Notice that one can consider an alternative definition of $\xi$ as $\xi^2 \equiv \frac{\sum_{x_2} |x_1 - x_2|^2 C_2(x_1, x_2)}{\sum_{x_2} C_2(x_1, x_2)}$ [1, 2]. With this definition $\xi$ is divergent on the Bethe lattice. This discrepancy is a pathology associated to the topology of the Bethe lattice in which the volume grows exponentially with the distance while it grows as a power law in finite dimension. In finite dimension this discrepancy is not present and indeed we will choose the second definition.

that obeys Eq. (4) with exponents

$$\sigma = \frac{1}{2}, \quad \text{and} \quad \tau = \frac{5}{2}, \tag{33}$$

that we identify with the mean-field values. In the next Sections we will consider percolation on the $M$-layer random lattice in finite dimension $D$. In the limit $M \to \infty$ the function $n(s,p)$ of the $M$-layer becomes identical to that of the Bethe lattice and therefore $\tau = 5/2$ and $\sigma = 1/2$. In addition, we will show that for $M \to \infty$ the two-point function obeys the scaling form (1) with exponents

$$\nu = \frac{1}{2}, \quad \eta = 0, \tag{34}$$

in all dimensions $D \geq 2$, see the comment after Eq. (64). Note that these relationships are consistent with (7) and (8) only for $D = D_U = 6$. Indeed $\tau = D/D_f + 1$ is a hyperscaling relationship that is not generically valid [22] at variance with the more general $\sigma^{-1} = \nu D_f$, which implies $D_f = 4$ for the $M \to \infty$ model in any dimension. Computing the $1/M$ corrections around the $M \to \infty$ limit, we will show that for $M$ *finite* the critical exponents are the same of the $M \to \infty$ limit for $D \geq D_U = 6$ while they are different for $D < D_U = 6$. On the other hand for $D < 6$ both relationships (7) and (8) hold. We note that the $M \to \infty$ model plays essentially the role of the Gaussian model in ferromagnetism, see [20], Chaps. 4 and 5.

# 4 The $M$-layer expansion

Conceptually the $M$-layer method is rather straightforward: 1) one introduces a $D$-dimensional random lattice depending on a parameter $M$, the limit $M \to \infty$ of the model is solvable as it coincides with the Bethe lattice solution; 2) then one computes the finite-$M$ corrections in powers of $1/M$ around the Bethe lattice solution. The goal is to study the critical behaviour near a second order phase transition for a model on a given lattice and, as we anticipated in Section 2, from the $1/M$ expansion one can obtain the $\epsilon$-expansion. The $M$-layer expansion has been introduced in Ref. [12] where diagrammatic rules were derived to compute $1/M$ corrections, in this Section we recall these rules, referring to the original paper for their derivation and all the details. Note that percolation itself is particularly useful to understand the origin of these rules and it is treated as an example in Section D of Ref. [12].

One can build the so-called $M$-layer construction considering $M$ different layers of the original model, and then rewiring the bonds between each nearest-neighboring node among the layers in such a way that each node on each layer still has the same number of neighbors, that now can be placed at different layers [24]. In the following we will focus on $D$-dimensional hyper-cubic lattices (for which the connectivity is $2D$), even if the $M$-layer construction can be applied to any type of lattice. We call "topological loop" a sequence of adjacent edges on the lattice that starts and ends in the same site. While finite dimensional lattices are characterized by the presence of many short topological loops, in the end of the procedure, the number of topological loops in the $M$-layer lattice will typically be reduced and in the $M \to \infty$ limit there will be no loops of finite length: the $M \to \infty$ solution of the model will correspond to the Bethe solution [19], computed on a random regular tree-like graph with the same fixed connectivity as the original model. At this point we can expand around this Bethe solution, introducing the small parameter $1/M$. The original model corresponds to $M = 1$, thus in principle one should need all orders in $1/M$ to obtain the correct solution for the original model. However, we are interested in the critical behaviour of the model, which should be independent of the actual value of $M$ due to universality. This expectation will indeed be confirmed in the context of percolation by the present computation. Furthermore, this implies that at each order in the

$1/M$ expansion we only need to consider the contributions that diverge the most approaching the critical point. One can show that the $1/M$ expansion for a generic $q$-point observable corresponds to an expansion in the number of topological loops considered when computing that observable. In the limit of large $M$, in a given realization of these random rewirings the $q$ sites considered for the observable will be connected with highest probability (proportional to $1/M$) by a sequence of adjacent edges (a "path") without topological loops, with lower probability by a path containing one topological loop and so on. In order to average over the rewirings the sum over all the possible realizations is needed, but we can retain the larger (in powers of $1/M$) contributions. We will call the path connecting the $q$ sites on a given realization a "topological diagram", that can contain an arbitrary number of topological loops, zero in the limit $M \to \infty$. In particular, if one wants to compute the $1/M$ expansion for a generic observable $O$, the following steps are required:

- Step 1: *Identify the possible topological diagrams*

  Depending on the order at which one wants to perform the expansion, one should identify the possible topological diagrams over which one needs to compute the chosen observable. If one is interested in the leading order, one should only look at diagrams without loops, that correspond to the Bethe locally tree-like topology. If one wants to compute the next-to-leading order, one has to identify all the possible topological diagrams that correspond to a Bethe lattice in which it has been manually injected a single topological loop, while any additional topological loop inserted will bring a new factor $1/M$ in the expansion.

- Step 2: *Weights, number of projections and symmetry factors*

  For any diagram $\mathcal{G}$ identified in Step 1, one needs to associate to it:

  - a weight $W(\mathcal{G})$, that will be a power of $1/M$ and will indicate the probability that a topological diagram of that kind is obtained in the rewiring procedure;

  - a symmetry factor $S(\mathcal{G})$, completely equivalent to that introduced in field theory for Feynman diagrams [21], that takes into account the number of ways in which vertices and lines can be switched leaving the topological structure of the diagram unaltered, see appendix C of Ref. [12] for a more detailed explanation of the equivalence between $S(\mathcal{G})$ and Feynman diagrams symmetry factors;

  - the number of realizations of the chosen topological diagram on the original lattice, $\mathcal{N}(\mathcal{G})$: just as an example, if the chosen diagram is a line of length $L$ between two points $x_1$ and $x_2$, the number of such diagrams in the $M$-layered lattice having a different projection on the original lattice corresponds to the number of non-backtracking paths (NBP) of length $L$ between the two points and its analytical expression is known in the literature [12, 25]. One can define $\mathcal{N}_L(x_1, x_2, \hat{\mu}, \hat{\nu})$ as the number of NBP of length $L$ where the directions $\hat{\mu}$ and $\hat{\nu}$ of the lines entering respectively in the external points $x_1$ and $x_2$ is fixed to one among the $2D$ possible ones. In the large $L$ limit, the actual value of the number of NBP will be independent on those directions, and we will simply define this number as $\mathcal{N}_L(x_1, x_2)$. The total number of the simple line diagrams of length $L$ between two points $x_1$ and $x_2$ will thus be $\mathcal{N}(\mathcal{G}) = (2D)^2 \mathcal{N}_L(x_1, x_2)$, where the factor $(2D)^2$ counts the possible entering directions of the line in the two external points. If one has a more complex diagram, to identify $\mathcal{N}(\mathcal{G})$ it is sufficient to multiply a factor $\mathcal{N}_L(x_i, x_j)$ for each internal line of length $L$, a factor $2D$ for each external vertex and a factor $\frac{(2D)!}{(2D-k)!}$ for any internal vertex of degree $k$, to count the different possible directions of the lines entering the vertex.

- Step 3: *Computation of the line-connected observable on the chosen diagram*

  For any diagram $\mathcal{G}$ identified in Step 1, one then needs to compute the value $O(\mathcal{G})$ of the chosen observable computed on a Bethe lattice in which the topological structure of that diagram has been manually injected. This observable will depend on the topology of the diagram and on the length of the lines. In order to compute, for a given observable, the expansion in the number of loops, or equivalently in powers of $1/M$, one should isolate different contributions coming from a given topological diagram. Generically, the contribution of a diagram without loops is contained in the one coming from the same diagram with some additional lines composing a loop. For this purpose we want to subtract the first contribution from the one coming from the loop diagram, this amounts to compute the so-called "line-connected observable", $O_{lc}(\mathcal{G})$. For the diagrams considered in this paper the following operative definition is sufficient: in order to compute $O_{lc}(\mathcal{G})$ one has to compute the observable on the given diagram $\mathcal{G}$ and then subtract all the contributions from the observable computed on diagrams where a line composing the loop (if any) is removed. For a more detailed treatment the reader is referred to Ref. [12].

- Step 4: *Sum of the contributions*

  At the end, we need to sum the contributions to the chosen observable coming from the different chosen diagrams. Because the values of the chosen observable only depend on the projection of the considered diagrams, for each diagram $\mathcal{G}$, we multiply the value of the line-connected observable $O_{lc}(\mathcal{G})$ by $\mathcal{N}(\mathcal{G})$, $S(\mathcal{G})$, $W(\mathcal{G})$, and we sum over the positions of internal vertices and over the lengths of the internal lines.

# 5  $M$-layer for percolation in $D$ dimensions

In this Section we apply the procedure described in the previous Section to the percolation problem. We consider both the problems of site and bond percolation on a hypercubic lattice in $D$ dimensions, which we denote $a_l \mathbb{Z}^D$, considering $a_l$ the lattice spacing. Following the notation of Sec. 2 we define $p$ (where $0 < p \leq 1$) as the probability that a site or an edge is present, for the case of site or bond percolation respectively. Since the $M$-layer approach is a way to construct an expansion for observables around the Bethe solution, we define the "bare mass"

$$\mu \equiv -\ln\left(\frac{p}{p_c}\right), \quad \text{for} \quad p \sim p_c, \tag{35}$$

where $p_c = 1/(2D-1)$ is the critical value for both site and bond percolation on a Bethe lattice with branching ratio $2D-1$, above which the so-called "giant cluster" is present.

Following the prescriptions of the $M$-layer construction [12,24] we report here the results of the application to both percolation problems in the non-percolating phase, $p < p_c$. We are interested in two observables: the two and three-point correlation functions $\overline{C_2(x_1, x_2)}$ and $\overline{C_3(x_1, x_2, x_3)}$, where $\overline{\cdot}$ denotes the average over the rewirings of the $M$-layer procedure. According to the $M$-layer rules these correlation functions will be written as sums, over different diagrams, of $\mathcal{C}_{n,lc}(\mathcal{G}; \{\mathcal{L}\})$, the $n$-point line-connected correlation, averaged over the realizations of the percolation problem and computed on the diagram $\mathcal{G}$, embedded on a tree graph, where $\{\mathcal{L}\}$ indicates the lengths of the different lines of the diagram. For both site and bond percolation, the two-point (three-point) correlation is defined as the probability that two (three) sites, at positions $x_1$ and $x_2$ ($x_1$, $x_2$ and $x_3$) are occupied and belong to the same cluster. In the end, at one loop level, we must subtract pieces already considered in loop-free diagrams, to compute the "line-connected" observable [12,24]. We analyse the two observables separately, following for each of them the steps listed in the previous Section.

$\mathcal{O}\left(\frac{1}{M}\right)$

$\mathcal{G}_1:$

$$\mathcal{O}\left(\frac{1}{M^2}\right) \quad \mathcal{G}_2: \qquad \mathcal{G}': \qquad \mathcal{G}'':$$

Figure 2: Diagrams that contribute to the two-point correlation functions up to one loop.

**Observable: $\overline{C_2(x_1, x_2)}$**

- Step 1: *Identify the possible topological diagrams*

  The simplest diagram connecting two points is the bare line, which we will call $\mathcal{G}_1$. Including the possibility of a loop to be present we consider the diagram composed of four lines with two vertices of degree three, where the two internal lines compose a loop. We will call this diagram $\mathcal{G}_2$.

  Other possibilities are the tadpole-type diagrams, connecting two points with a loop generated by one four-degree vertex or connecting two points by two three-degree vertices, respectively the diagrams $\mathcal{G}'$ and $\mathcal{G}''$ in Fig. 2. Nevertheless, these last two diagrams give no contributions to the *line-connected* two-point observable for percolation, as we will see in Step 3 below. We won't consider them in the following steps.

- Step 2: *Weights, number of projections and symmetry factors*

  - Diagram $\mathcal{G}_1$:
    * $W(\mathcal{G}_1) = \frac{1}{M}$;
    * $\mathcal{N}(\mathcal{G}_1; L; x_1, x_2) = (2D)^2 \mathcal{N}_L(x_1, x_2)$;
    * $S(\mathcal{G}_1) = 1$.
  - Diagram $\mathcal{G}_2$:
    * $W(\mathcal{G}_2) = \frac{1}{M^2}$;
    * $\mathcal{N}(\mathcal{G}_2; \vec{L}; x_1, x_2) = (2D)^2 \left(\frac{(2D)!}{(2D-3)!}\right)^2 \sum_{x_0, x_0'} \mathcal{N}_{L_1}(x_1, x_0) \mathcal{N}_{L_2}(x_0', x_2) \prod_{i=A,B} \mathcal{N}_{L_i}(x_0, x_0')$;
    * $S(\mathcal{G}_2) = 2$,

    where $\vec{L} = (L_1, L_A, L_B, L_2)$.

- Step 3: *Computation of $\mathcal{C}_{2,lc}(\mathcal{G}_1; L)$ and $\mathcal{C}_{2,lc}(\mathcal{G}_2; \vec{L})$*

  Given the definition of the line-connected two-point correlation for both percolation problems, we firstly compute the contributions of diagrams $\mathcal{G}_1$ and $\mathcal{G}_2$ for the problem of site percolation:

$$\mathcal{C}_{2,lc}(\mathcal{G}_1; L) = p\, p^L, \tag{36}$$

$$\mathcal{C}_{2,lc}(\mathcal{G}_2; \vec{L}) = -p^{L_1+L_2+L_A+L_B}. \tag{37}$$

The first result is immediate since, in the non-percolating phase, all the $L+1$ sites, connected by a line of length $L$, must be active, in order to connect the two sites at the extremities. The second result appears because, for the sites at the extremities to be

connected, one or both lines of the loop must consist on active sites, in addition to the external lines, which also need to be composed of active sites. The associated probability for this to happen is $p^{L_1+1}(p^{L_A-1} + p^{L_B-1} - p^{L_A+L_B-2})p^{L_2+1}$. The aforementioned result is obtained subtracting the straight line contributions, already taken into account with $\mathcal{G}_1$: $p^{L_1+1}p^{L_A-1}p^{L_2+1}$ and $p^{L_1+1}p^{L_B-1}p^{L_2+1}$. This last operation is the application of the "line-connected" definition [12].

Performing the same computation for diagrams $\mathcal{G}'$ and $\mathcal{G}''$ we obtain zero, as anticipated. The reason is that the two tadpoles, that enter the site $x_0$, do not change the probability that sites $x_1$ and $x_2$ belong to the same cluster with respect to the case where the loop is not present. Indeed, independently of the lines of the tadpole, site $x_0$ must be active in order to connect the two sites, then, subtracting the contributions needed to define the line-connected observable, that are the simple lines without tadpoles, the net contribution is zero. These diagrams are instead relevant in the percolating phase that we aim to study in a subsequent work.

A similar computation can be performed for the bond percolation. In this case, considering the contribution of $\mathcal{G}_1$, in order for the two sites to be in the same cluster, all the edges connecting the two must be active:

$$\mathcal{C}_{2,lc}^{\text{bond}}(\mathcal{G}_1; L) = p^L \,, \tag{38}$$

similarly, the line-connected contribution for $\mathcal{G}_2$, is

$$\mathcal{C}_{2,lc}^{\text{bond}}(\mathcal{G}_2; \vec{L}) = -p^{L_1+L_2+L_A+L_B} \,. \tag{39}$$

The same argument, used for the site percolation problem, can be applied to the topological diagrams $\mathcal{G}'$ and $\mathcal{G}''$ in the bond percolation case, for which they give zero contribution too.

- Step 4: *Sum of the contributions*

The expression for $\overline{C_2(x_1, x_2)}$, that is for the site percolation, is

$$\begin{aligned}
\overline{C_2(x_1, x_2)} = {}&\frac{1}{M} \sum_L \mathcal{N}(\mathcal{G}_1; L; x_1, x_2) \mathcal{C}_{2,lc}(\mathcal{G}_1; L) \\
&+ \frac{1}{2M^2} \sum_{\vec{L}} \mathcal{N}(\mathcal{G}_2; \vec{L}; x_1, x_2) \mathcal{C}_{2,lc}(\mathcal{G}_2; \vec{L}) + \mathcal{O}\left(\frac{1}{M^3}\right),
\end{aligned} \tag{40}$$

while, for the bond percolation problem, we have

$$\begin{aligned}
\overline{C_2^{\text{bond}}(x_1, x_2)} = {}&\frac{1}{M} \sum_L \mathcal{N}(\mathcal{G}_1; L; x_1, x_2) \mathcal{C}_{2,lc}^{\text{bond}}(\mathcal{G}_1; L) \\
&+ \frac{1}{2M^2} \sum_{\vec{L}} \mathcal{N}(\mathcal{G}_2; \vec{L}; x_1, x_2) \mathcal{C}_{2,lc}^{\text{bond}}(\mathcal{G}_2; \vec{L}) + \mathcal{O}\left(\frac{1}{M^3}\right).
\end{aligned} \tag{41}$$

We can notice that the only difference is for the observable computed on a given diagram, here $\mathcal{G}_1$ and $\mathcal{G}_2$, which is the only model dependent part of the $M$-layer computations.

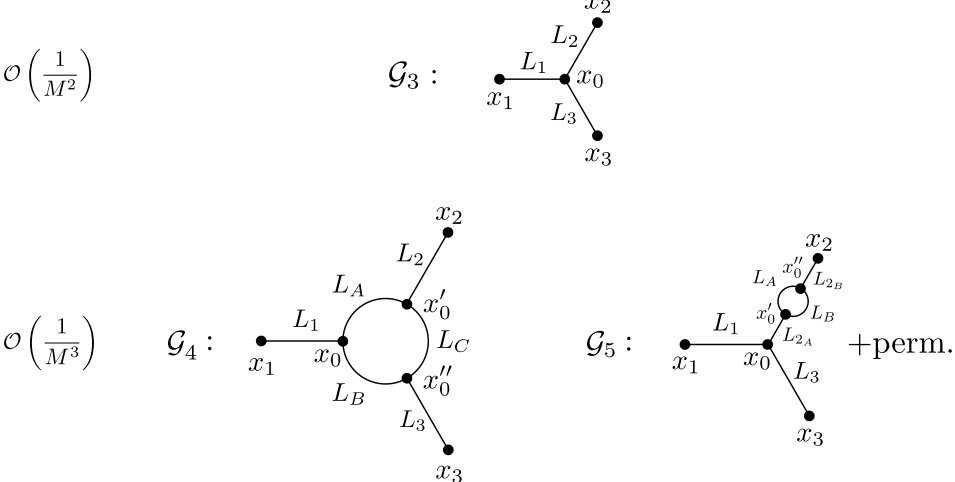

Figure 3: Diagrams that contribute to the three-point correlation functions up to one loop.

**Observable:** $\overline{C_3(x_1, x_2, x_3)}$

- Step 1: *Identify the possible topological diagrams*

  The simplest diagram connecting three points is the bare three-degree vertex, which we will call $\mathcal{G}_3$. Including the possibility for a loop to be present, we consider the diagram, composed of six lines, with three vertices of degree three, we will call this diagram $\mathcal{G}_4$. At one-loop level there are three more diagrams connecting three points with a single loop, which are the same as $\mathcal{G}_3$, but where one of the external legs is dressed with $\mathcal{G}_2$. We call such a diagram $\mathcal{G}_5$, including all the permutations. All these diagrams are reported in Fig. 3.

- Step 2: *Weights, number of projections and symmetry factors*

  - Diagram $\mathcal{G}_3$:
    * $W(\mathcal{G}_3) = \frac{1}{M^2}$;
    * $\mathcal{N}(\mathcal{G}_3; \vec{L}'; x_1, x_2, x_3) = (2D)^3 \frac{(2D)!}{(2D-3)!} \sum_{x_0} \prod_{i=i}^3 \mathcal{N}_{L_i}(x_i, x_0)$;
    * $S(\mathcal{G}_3) = 1$.

  - Diagram $\mathcal{G}_4$:
    * $W(\mathcal{G}_4) = \frac{1}{M^3}$;
    * $\mathcal{N}(\mathcal{G}_4; \vec{L}''; x_1, x_2, x_3) = (2D)^3 \left(\frac{(2D)!}{(2D-3)!}\right)^3$
      $\times \sum_{x_0, x_0', x_0''} \mathcal{N}_{L_1}(x_1, x_0) \mathcal{N}_{L_2}(x_2, x_0') \mathcal{N}_{L_3}(x_3, x_0'') \mathcal{N}_{L_A}(x_0, x_0') \mathcal{N}_{L_B}(x_0, x_0'') \mathcal{N}_{L_C}(x_0', x_0'')$;
    * $S(\mathcal{G}_4) = 1$.

  - Diagram $\mathcal{G}_5$:
    * $W(\mathcal{G}_5) = \frac{1}{M^3}$;
    * $\mathcal{N}(\mathcal{G}_5; \vec{L}'''; x_1, x_2, x_3) = (2D)^3 \left(\frac{(2D)!}{(2D-3)!}\right)^3$
      $\times \sum_{x_0, x_0', x_0''} \mathcal{N}_{L_1}(x_1, x_0) \mathcal{N}_{L_{2_A}}(x_0, x_0') \mathcal{N}_{L_{2_B}}(x_2, x_0'') \mathcal{N}_{L_3}(x_3, x_0) \prod_{i=A,B} \mathcal{N}_{L_i}(x_0', x_0'')$;
    * $S(\mathcal{G}_5) = 2$,

    where $\vec{L}' = (L_1, L_2, L_3)$, $\vec{L}'' = (\vec{L}', L_A, L_B, L_C)$ and $\vec{L}''' = (L_1, L_{2_A}, L_A, L_B, L_{2_B}, L_3)$.

- Step 3: *Computation of* $\mathcal{C}_{3,lc}(\mathcal{G}_3;\vec{L}')$, $\mathcal{C}_{3,lc}(\mathcal{G}_4;\vec{L}'')$ *and* $\mathcal{C}_{3,lc}(\mathcal{G}_5;\vec{L}''')$

  As for the two-point function we compute the contributions, starting from the site percolation problem:

$$\mathcal{C}_{3,lc}(\mathcal{G}_3;\vec{L}') = p\, p^{L_1+L_2+L_3}\,, \tag{42}$$

$$\mathcal{C}_{3,lc}(\mathcal{G}_4;\vec{L}'') = -2p^{L_1+L_2+L_3+L_A+L_B+L_C}\,, \tag{43}$$

$$\mathcal{C}_{3,lc}(\mathcal{G}_5;\vec{L}''') = -p^{L_1+L_{2_A}+L_{2_B}+L_A+L_B+L_3}\,. \tag{44}$$

  The result for $\mathcal{G}_3$ is easily derived, considering that all the sites of the topology must be active for the extremities to be connected. The result for $\mathcal{G}_5$ is obtained by multiplying the contribution for the bare vertex by the loop correction of the two-point function, diagram $\mathcal{G}_2$, with the corresponding lengths. The contribution of $\mathcal{G}_4$ is a generalization of the computation for $\mathcal{G}_2$; to connect the three extremities two of the three (or all the three) lines of the loop must consist on all active sites. Moreover, in this case we have to subtract three contributions, corresponding to cutting $L_A$, $L_B$, and $L_C$ respectively, already included in the bare contribution $\mathcal{G}_3$.

  Analogously to the two-point function, we use the same arguments to compute the contributions for the bond percolation three-point function:

$$\mathcal{C}_{3,lc}^{bond}(\mathcal{G}_3;\vec{L}') = p^{L_1+L_2+L_3}\,, \tag{45}$$

$$\mathcal{C}_{3,lc}^{bond}(\mathcal{G}_4;\vec{L}'') = -2p^{L_1+L_2+L_3+L_A+L_B+L_C}\,, \tag{46}$$

$$\mathcal{C}_{3,lc}^{bond}(\mathcal{G}_5;\vec{L}''') = -p^{L_1+L_{2_A}+L_{2_B}+L_A+L_B+L_3}\,. \tag{47}$$

- Step 4: *Sum of the contributions*

  The expression for $\overline{C_3(x_1,x_2,x_3)}$, that is for the site percolation, is

$$\overline{C_3(x_1,x_2,x_3)} = \frac{1}{M^2}\sum_{\vec{L}'}\mathcal{N}(\mathcal{G}_3;\vec{L}';x_1,x_2,x_3)\mathcal{C}_{3,lc}(\mathcal{G}_3;\vec{L}')$$

$$+\frac{1}{M^3}\sum_{\vec{L}''}\mathcal{N}(\mathcal{G}_4;\vec{L}'';x_1,x_2,x_3)\mathcal{C}_{3,lc}(\mathcal{G}_4;\vec{L}'')$$

$$+\frac{1}{2M^3}\sum_{\vec{L}'''}\mathcal{N}(\mathcal{G}_5;\vec{L}''';x_1,x_2,x_3)\mathcal{C}_{3,lc}(\mathcal{G}_5;\vec{L}''')+\mathcal{O}\left(\frac{1}{M^4}\right). \tag{48}$$

As noticed for the two-point function, the expression of $\overline{C_3^{bond}(x_1,x_2,x_3)}$, that is for the bond percolation, is the same as $\overline{C_3(x_1,x_2,x_3)}$ with the corresponding observables: $\mathcal{C}_{3,lc}^{bond}(\mathcal{G}_3;\vec{L}')$, $\mathcal{C}_{3,lc}^{bond}(\mathcal{G}_4;\vec{L}'')$ and $\mathcal{C}_{3,lc}^{bond}(\mathcal{G}_5;\vec{L}''')$. We do not write it for brevity.

In appendix C we discuss why we didn't include other possible but *irrelevant* diagrams to study the critical behavior of the percolation problem and in appendix D we present the explicit computation of the leading order critical behaviour of the four-point correlation function.

**Computation of the moments of** $n(s,p)$  In order to compute $\chi_2$ and $\chi_3$ we Fourier transform $\overline{C_2(x_1,x_2)}$ and $\overline{C_3(x_1,x_2,x_3)}$, given in Eqs. (40) and (48), using the following convention:

$$\widehat{h}(k) = a_l^D\sum_{x\in a_l\mathbb{Z}^D}h(x)e^{ikx}\,,\quad h(x) = \int_{\left[-\frac{\pi}{a_l},\frac{\pi}{a_l}\right]}\frac{d^Dk}{(2\pi)^D}\widehat{h}(k)e^{-ikx}\,, \tag{49}$$

that implies

$$\left(\frac{2\pi}{a_l}\right)^D \delta^D(k) = \sum_{x \in a_l \mathbb{Z}^D} e^{ikx}.$$ (50)

We also use the fact that $\mathcal{N}_L(x_1, x_2)$ is a function of the difference between the starting and arrival point only, so that, in Fourier space, we have

$$\widehat{\mathcal{N}}_L(k_1, k_2) = (2\pi)^D \delta^D(k_1 + k_2) \widehat{\mathcal{N}}_L(k_1),$$ (51)

where, for small $k$ [12, 15],

$$\widehat{\mathcal{N}}_L(k) \approx 2D(2D-1)^{L-1} a_l^D e^{-k^2 a_l^2 L/(2D-2)}.$$ (52)

In view of the fact that in the critical region the sums will be dominated by large $L$ contributions, we may write the sums over the lengths as integrals:

$$\sum_{L=1}^{\infty} \rightarrow \int_{1/\Lambda^2}^{\infty} dL,$$ (53)

where we introduced the UV cutoff $\Lambda = 1$ to make contact with field theory. Note that while in field-theory the UV cutoff is inserted manually, in the $M$-layer construction it arises naturally due to the lattice spacing (see more details in appendix B). The resulting expressions, for the site percolation case, of the two and three-point functions are respectively

$$\overline{\widehat{C}_2(k, k')} = \frac{\widehat{C}\widehat{B}^2 a_l^D}{\widehat{A}\mu} \frac{1}{\widehat{k}^2 + 1}(2\pi)^D \delta^D(k + k')$$

$$\times \left(1 - \frac{\widehat{A}\mu^{\frac{D}{2}-3}}{2(\widehat{k}^2 + 1)} \int \frac{d^D\widehat{q}}{(2\pi)^D} \int d\widehat{L}_A d\widehat{L}_B e^{-(1+(\widehat{k}-\widehat{q})^2)\widehat{L}_A} e^{-(1+\widehat{q}^2)\widehat{L}_B}\right) + \mathcal{O}\left(\frac{1}{M^3}\right),$$ (54)

and

$$\overline{\widehat{C}_3(k_1, k_2, k_3)} = \frac{\widehat{C}\widehat{B}^3 a_l^{2D}}{\widehat{A}\mu^3} \frac{(2\pi)^D \delta^D(k_1 + k_2 + k_3)}{(\widehat{k}_1^2 + 1)(\widehat{k}_2^2 + 1)(\widehat{k}_3^2 + 1)}$$ (55)

$$\times \left(1 - 2\widehat{A}\mu^{\frac{D}{2}-3} \int \frac{d^D\widehat{q}}{(2\pi)^D} \int d\widehat{L}_A d\widehat{L}_B d\widehat{L}_C \, e^{-(1+(\widehat{k}_2+\widehat{k}_3+\widehat{q})^2)\widehat{L}_A} e^{-(1+(\widehat{k}_2+\widehat{q})^2)\widehat{L}_B} e^{-(1+\widehat{q}^2)\widehat{L}_C}\right.$$

$$\left. - \frac{1}{2}\frac{\widehat{A}\mu^{\frac{D}{2}-3}}{(\widehat{k}_2+\widehat{k}_3)^2 + 1} \int \frac{d^D\widehat{q}}{(2\pi)^D} \int d\widehat{L}_A d\widehat{L}_B \, e^{-(1+(\widehat{k}_2+\widehat{q})^2)\widehat{L}_A} e^{-(1+\widehat{q}^2)\widehat{L}_B} + \text{perm.}\right) + \mathcal{O}\left(\frac{1}{M^4}\right),$$

where $\mu$ is the one defined in Eq. (35). We also defined the following non-universal constants:

$$\widehat{A} \equiv \frac{1}{M}\left(\frac{(2D)!}{(2D-3)!}\right)^2 p^{-1}(2D-2)^{\frac{D}{2}}\left(\frac{2D}{2D-1}\right)^3,$$ (56)

$$\widehat{B} \equiv \frac{1}{M}2D\left(\frac{(2D)!}{(2D-3)!}\right)\left(\frac{2D}{2D-1}\right)^2,$$ (57)

$$\widehat{C} \equiv (2D-2)^{\frac{D}{2}},$$ (58)

and we rescaled the momenta and lengths according to:

$$\widehat{k} \equiv k\frac{a_l}{\sqrt{\mu(2D-2)}}, \quad \text{and} \quad \widehat{L}_i \equiv L_i \mu.$$ (59)

Note that in Eqs. (54), (55) we have omitted the the extremes of integration $(\mu/\Lambda^2, \infty)$ of the integrals over $\widehat{L}$. In appendix A we show how to generalize this kind of computation for a $V_e$-point function, with $V_e \geq 2$, moreover we explain the reasoning behind the identification of the constants $\widehat{A}$, $\widehat{B}$ and $\widehat{C}$. The same steps can be done for the bond percolation problem, the only difference being the definition of the non-universal constant $\widehat{A}$:

$$\widehat{A}_{bond} \equiv \frac{1}{M}\left(\frac{(2D)!}{(2D-3)!}\right)^2 (2D-2)^{\frac{D}{2}}\left(\frac{2D}{2D-1}\right)^3, \tag{60}$$

in which no factor $p^{-1}$ appears, at variance with Eq. (56). In the following we will perform explicit computations for the site problem only, the reader can reproduce them for the bond percolation simply using Eq. (60) instead of Eq. (56).

In appendix B we show that the above expression, for $\overline{\widehat{C}_2(k,k')}$ and $\overline{\widehat{C}_3(k_1,k_2,k_3)}$, are precisely the same that appear from the Feynman diagrams of the corresponding scalar cubic field-theory obtained from the Fortuin-Kasteleyn mapping to the $n+1$-state Potts model in the limit $n \to 0$, corresponding to percolation [6–9].

From the above expressions we compute the functions $\chi_q$ introduced in Section 2. Notice that we did not rescale the momenta inside the momentum conservation delta functions, thus, to compute $\chi_q$, according to Eq. (5), we have simply to divide by $a_l^{(q-1)D}$, remove $(2\pi)^D$ times the conservation delta function and set the external momenta to zero. This leads to

$$\chi_2(\mu) = \frac{\widehat{C}\widehat{B}^2}{\widehat{A}\mu}\left(1 - \frac{\widehat{A}\mu^{\frac{D}{2}-3}}{2(4\pi)^{\frac{D}{2}}}\int \frac{d\widehat{L}_A d\widehat{L}_B}{(\widehat{L}_A+\widehat{L}_B)^{\frac{D}{2}}}e^{-\widehat{L}_A-\widehat{L}_B}\right) + \mathcal{O}\left(\frac{1}{M^3}\right), \tag{61}$$

$$\chi_3(\mu) = \frac{\widehat{C}\widehat{B}^3}{\widehat{A}\mu^3}\left(1 - \frac{2\widehat{A}\mu^{\frac{D}{2}-3}}{(4\pi)^{\frac{D}{2}}}\int \frac{d\widehat{L}_A d\widehat{L}_B d\widehat{L}_C}{(\widehat{L}_A+\widehat{L}_B+\widehat{L}_C)^{\frac{D}{2}}}e^{-\widehat{L}_A-\widehat{L}_B-\widehat{L}_C}\right.$$
$$\left.- \frac{3}{2}\widehat{A}\mu^{\frac{D}{2}-3}\int \frac{d\widehat{L}_A d\widehat{L}_B}{(\widehat{L}_A+\widehat{L}_B)^{\frac{D}{2}}}e^{-\widehat{L}_A-\widehat{L}_B}\right) + \mathcal{O}\left(\frac{1}{M^4}\right). \tag{62}$$

**Ginzburg criterion for $D_U$**    Once the two-point function is computed, in this paper using the $M$-layer construction, it is possible to deduce the upper critical dimension of the problem, $D_U$, applying the Ginzburg criterion in the non-critical phase [26]. We first introduce the function $\widehat{G}(k)$, corresponding to the *propagator* in the field-theoretical language, as

$$\overline{\widehat{C}_2(k,k')} \equiv (2\pi)^D \delta^D(k+k')\widehat{G}(k). \tag{63}$$

Using this defintion, together with Eq. (54), we have

$$\widehat{G}(k) \propto \frac{1}{M}\frac{1}{\rho k^2+\mu}\left(1 - \frac{1}{M}\frac{c}{\rho k^2+\mu}\int \frac{d^D q}{(2\pi)^D}\sum_{L_A,L_B=1}^{\infty}e^{-(\rho(k-q)^2+\mu)L_A}e^{-(\rho q^2+\mu)L_B}\right) + \mathcal{O}\left(\frac{1}{M^3}\right), \tag{64}$$

where we rescaled momenta and lengths according to Eq. (59), with $\rho \equiv a_l^2/(2D-2)$ and $c \equiv M\widehat{A}/2$ defined in order to make the $1/M$ factors explicit. Notice that we also made use of the relation in Eq. (53) to write sums instead of integrals. Here we understand that for $M \to \infty$ the correction can be neglected and the two-point function assumes the mean-field expression, *i.e.* the Gaussian propagator, which leads to the mean-field values for the anomalous dimension, $\eta = 0$, and the exponent associated to the correlation length, $\nu = 1/2$. Moreover, in high dimensions we expect for the two-point function the following Gaussian form near the critical point and for $k^2 \to 0$,

$$\left(M\widehat{G}(k)\right)^{-1} \propto \mathcal{A}(\mu-\mu_c) + \mathcal{B}\rho k^2 + \mathcal{O}\left(k^4\right), \tag{65}$$

from which we have the correction, at order $1/M$, to the control parameter

$$\mu_c = -\frac{c}{M} \frac{1}{(4\pi\rho)^{\frac{D}{2}}} \sum_{L_A, L_B=1}^{\infty} \frac{1}{(L_A + L_B)^{\frac{D}{2}}}, \tag{66}$$

and the two prefactors

$$\mathcal{A} = 1 - \frac{c}{M} \frac{1}{(4\pi\rho)^{\frac{D}{2}}} \sum_{L_A, L_B=1}^{\infty} \frac{1}{(L_A + L_B)^{\frac{D}{2}-1}}, \tag{67}$$

$$\mathcal{B} = 1 - \frac{c}{M} \frac{1}{(4\pi\rho)^{\frac{D}{2}}} \sum_{L_A, L_B=1}^{\infty} \frac{L_A L_B}{(L_A + L_B)^{\frac{D}{2}+1}}. \tag{68}$$

We notice that these three corrections, $\mu_c$, $\mathcal{A}$ and $\mathcal{B}$, diverge respectively for $D \leq 4$, $D \leq 6$ and $D \leq 6$, revealing that the upper critical dimension for the site percolation problem, where the mean-field behavior breaks down, is $D_U = 6$. Again we notice that the analysis doesn't change considering bond percolation, since the only difference is in the definition of the factor c, not relevant for these divergences. In order to go below the upper critical dimension we can rewrite the propagator, including the cutoff in the integrals as prescribed by Eq. (53):

$$\left(M\widehat{G}_2(k)\right)^{-1} \propto \mu(\widehat{k}^2 + 1)\left(1 + \frac{c}{M}\frac{\mu^{\frac{D}{2}-3}}{(\widehat{k}^2 + 1)}\frac{1}{(4\pi)^{D/2}}\int_{\mu/\Lambda^2}^{\infty} d\widehat{L}_A d\widehat{L}_B \frac{1}{(L_A + L_B)^{\frac{D}{2}}} e^{-\widehat{L}_A - \widehat{L}_B - \widehat{k}^2 \frac{L_A L_B}{L_A + L_B}}\right)$$
$$+ \mathcal{O}\left(\frac{1}{M^3}\right). \tag{69}$$

We understand that the correction is not negligible for $D < 6$ in the limit $\mu \to 0$, due to the presence of the term $\mu^{\frac{D}{2}-3}$. Moreover the integrals over $L_A$ and $L_B$ diverge in the ultraviolet (UV) regime, that is for $\mu/\Lambda^2 \to 0$ if $D \geq 4$ and in particular for $D \simeq 6$ from below. In order for the integrals to be finite in the limit $\mu/\Lambda^2 \to 0$ we should perform the standard mass renormalization, changing variable from $\mu$ to $m^2 \equiv \xi^{-2}$, to be explicitly done in the next Section.

## 6 Computation of critical exponents

In this Section we start from the results of the $M$-layer construction for the two and three-point observables and we perform standard procedures in order to compute the $\epsilon$-expansion for the critical exponents. From the definition of $\widehat{G}(k)$, Eq. (63) we can define the correlation length $\xi$:

$$\xi^2 \equiv \widehat{G}(0)\frac{\partial \widehat{G}^{-1}(k)}{\partial k^2}\bigg|_{k^2=0}, \tag{70}$$

where, with a little abuse of notation, we identify with $k$ the modulus of the corresponding vector. Since

$$\frac{\partial}{\partial k^2} = \frac{\partial \widehat{k}^2}{\partial k^2}\frac{\partial}{\partial \widehat{k}^2} = \frac{a_l^2}{\mu \widehat{C}^{\frac{2}{D}}}\frac{\partial}{\partial \widehat{k}^2}, \tag{71}$$

we have:

$$\widehat{G}(0) = \frac{\widehat{C}\widehat{B}^2 a_l^D}{\widehat{A}\mu}\left(1 - \frac{\widehat{A}\mu^{\frac{D}{2}-3}}{2(4\pi)^{\frac{D}{2}}}\int \frac{d\widehat{L}_A d\widehat{L}_B}{(\widehat{L}_A + \widehat{L}_B)^{\frac{D}{2}}}e^{-\widehat{L}_A - \widehat{L}_B}\right), \tag{72}$$

and for small $\widehat{A}$ (that is for large $M$):

$$\widehat{G}^{-1}(k) \simeq \frac{\widehat{A}\mu}{\widehat{C}\widehat{B}^2 a_l^D} \left( \widehat{k}^2 + 1 + \frac{\widehat{A}\mu^{\frac{D}{2}-3}}{2(4\pi)^{\frac{D}{2}}} \int \frac{d\widehat{L}_a d\widehat{L}_b}{(\widehat{L}_a + \widehat{L}_b)^{\frac{D}{2}}} e^{-\frac{\widehat{L}_a \widehat{L}_b}{\widehat{L}_a + \widehat{L}_b}\widehat{k}^2 - \widehat{L}_a - \widehat{L}_b} \right), \qquad (73)$$

where in the r.h.s. we have replaced $k$ with $\hat{k}$ according to the definition given in (59). We then obtain:

$$\left. \frac{\partial \widehat{G}^{-1}(k)}{\partial \widehat{k}^2} \right|_{\widehat{k}^2=0} = \frac{\widehat{A}\mu}{\widehat{C}\widehat{B}^2 a_l^D} \left( 1 + \frac{\widehat{A}\mu^{\frac{D}{2}-3}}{2(4\pi)^{\frac{D}{2}}} \int \frac{d\widehat{L}_a d\widehat{L}_b}{(\widehat{L}_a + \widehat{L}_b)^{\frac{D}{2}}} e^{-\widehat{L}_a - \widehat{L}_b} \left. \frac{\partial}{\partial \widehat{k}^2} \left( e^{-\frac{\widehat{L}_a \widehat{L}_b}{\widehat{L}_a + \widehat{L}_b}\widehat{k}^2} \right) \right|_{\widehat{k}^2=0} \right), \quad (74)$$

where

$$\int \frac{d\widehat{L}_a d\widehat{L}_b}{(\widehat{L}_a + \widehat{L}_b)^{\frac{D}{2}}} e^{-\widehat{L}_a - \widehat{L}_b} \left. \frac{\partial}{\partial \widehat{k}^2} \left( e^{-\frac{\widehat{L}_a \widehat{L}_b}{\widehat{L}_a + \widehat{L}_b}\widehat{k}^2} \right) \right|_{\widehat{k}^2=0} = -\int \frac{d\widehat{L}_a d\widehat{L}_b}{(\widehat{L}_a + \widehat{L}_b)^{\frac{D}{2}+1}} \widehat{L}_A \widehat{L}_B e^{-\widehat{L}_a - \widehat{L}_b}. \qquad (75)$$

We want to notice that in Eqs. (72), (73) and (74) we neglected higher orders, with respect to the one-loop corrections, in powers of $1/M$. From now on we will neglect these terms if not explicitly specified. Defining

$$I_\alpha(\mu) \equiv \int_{\mu/\Lambda^2}^\infty d\widehat{L}_a d\widehat{L}_b \frac{e^{-\widehat{L}_a - \widehat{L}_b}}{(\widehat{L}_a + \widehat{L}_b)^{\frac{D}{2}}}, \qquad (76)$$

and

$$I_\beta(\mu) \equiv \int_{\mu/\Lambda^2}^\infty d\widehat{L}_a d\widehat{L}_b \frac{\widehat{L}_a \widehat{L}_b}{(\widehat{L}_a + \widehat{L}_b)^{\frac{D}{2}+1}} e^{-\widehat{L}_a - \widehat{L}_b}, \qquad (77)$$

we have

$$\xi^2(\mu) = \frac{1}{m^2(\mu)} = \frac{a_l^2}{\widehat{C}^{\frac{2}{D}}\mu} \left( 1 - \frac{1}{2} \frac{\widehat{A}\mu^{\frac{D}{2}-3}}{(4\pi)^{\frac{D}{2}}} \left( I_\alpha(\mu) + I_\beta(\mu) \right) \right). \qquad (78)$$

In the integrals in Eqs. (76), (77), we have written explicitly the extremes of integration that we have omitted previously. Notice that the integral $I_\alpha(\mu)$ is UV divergent in $D=6$ for $\mu \to 0$ (i.e., $p \to p_c$). Now we can simply invert the relation, to express $\mu$ as a function of $m^2$:

$$\mu(m^2) = a_l^2 \widehat{C}^{-\frac{2}{D}} m^2 \left( 1 - \frac{1}{2} \frac{\widehat{A}m^{D-6}\widehat{C}^{\frac{6}{D}-1} a_l^{D-6}}{(4\pi)^{\frac{D}{2}}} \left( I_\alpha\left(\mu(m^2)\right) + I_\beta\left(\mu(m^2)\right) \right) \right). \qquad (79)$$

Notice that the previous equations for $\chi_2$ and $\chi_3$ are written as functions of $\mu$, which is not the "physical mass", thus they can be divergent, for $\mu \to 0$, near the upper critical dimension, $D_U = 6$. To avoid the divergences we need the expression of $\mu$ as a function of $m^2$, to correctly write $\lambda$, as defined in Eq. (83). To this aim we compute $\xi^2(\mu)$ (and so $m^2(\mu)$) from its definition.

At this point we have all the ingredients to write $\chi_2$ and $\chi_3$ as functions of the physical parameter $m^2$. Plugging Eq. (79) into Eqs. (61) and (62) we obtain:

$$\begin{aligned}
\chi_2(m^2) &= \frac{\widehat{C}\widehat{B}^2\widehat{C}^{\frac{2}{D}}}{\widehat{A}a_l^2} m^{-2} \left( 1 + \frac{1}{2} \frac{\widehat{A}m^{D-6}\widehat{C}^{\frac{6}{D}-1} a_l^{D-6}}{(4\pi)^{\frac{D}{2}}} \left( I_\alpha\left(\mu(m^2)\right) + I_\beta\left(\mu(m^2)\right) \right) \right) \\
&\quad \times \left( 1 - \frac{1}{2} \frac{\widehat{A}m^{D-6}\widehat{C}^{\frac{6}{D}-1} a_l^{D-6}}{(4\pi)^{\frac{D}{2}}} I_\alpha\left(\mu(m^2)\right) \right) \\
&= \frac{\widehat{C}\widehat{B}^2\widehat{C}^{\frac{2}{D}}}{\widehat{A}a_l^2} m^{-2} \left( 1 + \frac{1}{2} \frac{\widehat{A}m^{D-6}\widehat{C}^{\frac{6}{D}-1} a_l^{D-6}}{(4\pi)^{\frac{D}{2}}} I_\beta\left(\mu(m^2)\right) \right),
\end{aligned} \qquad (80)$$

$$\chi_3\left(m^2\right) = \frac{\widehat{C}\widehat{B}^3\widehat{C}^{\frac{6}{D}}}{\widehat{A}a_l^6}m^{-6}\left(1 + \frac{3}{2}\frac{\widehat{A}m^{D-6}\widehat{C}^{\frac{6}{D}-1}a_l^{D-6}}{(4\pi)^{\frac{D}{2}}}\left(I_\alpha\left(\mu(m^2)\right) + I_\beta\left(\mu(m^2)\right)\right)\right)$$

$$\times\left(1 - 2\frac{\widehat{A}m^{D-6}\widehat{C}^{\frac{6}{D}-1}a_l^{D-6}}{(4\pi)^{\frac{D}{2}}}I_\gamma\left(\mu(m^2)\right) - \frac{3}{2}\frac{\widehat{A}m^{D-6}\widehat{C}^{\frac{6}{D}-1}a_l^{D-6}}{(4\pi)^{\frac{D}{2}}}I_\alpha\left(\mu(m^2)\right)\right)$$

$$= \frac{\widehat{C}\widehat{B}^3\widehat{C}^{\frac{6}{D}}}{\widehat{A}a_l^6}m^{-6}\left(1 + \frac{\widehat{A}m^{D-6}\widehat{C}^{\frac{6}{D}-1}a_l^{D-6}}{(4\pi)^{\frac{D}{2}}}\left(\frac{3}{2}I_\beta\left(\mu(m^2)\right) - 2I_\gamma\left(\mu(m^2)\right)\right)\right), \qquad (81)$$

where

$$I_\gamma(\mu) \equiv \int_{\mu/\Lambda^2}^{\infty} d\widehat{L}_A d\widehat{L}_B d\widehat{L}_C \frac{e^{-\widehat{L}_A-\widehat{L}_B-\widehat{L}_C}}{(\widehat{L}_A + \widehat{L}_B + \widehat{L}_C)^{\frac{D}{2}}}. \qquad (82)$$

Notice that $\chi_2(\mu)$ and $\chi_3(\mu)$ have UV divergences near 6 dimensions due the presence of $I_\alpha(\mu)$, which disappears when they are written as functions of $m$, i.e. $\chi_2\left(m^2\right)$ and $\chi_3\left(m^2\right)$ are free of UV divergences near 6 dimensions.

**Critical exponents in fixed dimension**     In this Section we perform the fixed-dimension computation of the critical exponents [20]. Led by the scaling laws discussed in Sec. 2, we compute the following dimensionless ratio:

$$\lambda \equiv \left(\frac{a_l}{\xi}\right)^D \frac{\chi_3^2(m^2)}{\chi_2^3(m^2)}. \qquad (83)$$

On the other hand $m^2$ is connected to the bare distance from the critical point by

$$m^2 \sim |\mu - \mu_c|^{2\nu}, \quad \text{and} \quad \xi \sim |\mu - \mu_c|^{-\nu}, \qquad (84)$$

where $\nu$ is the critical exponent for the divergence of the correlation length. In the end, defining

$$u \equiv \widehat{A}\widehat{C}^{\frac{6}{D}-1}a_l^{D-6}m^{D-6} \equiv g\, m^{D-6}, \qquad (85)$$

we can compute the ratio $\lambda$

$$\lambda = u\left(1 - 2\frac{u}{(4\pi)^{\frac{D}{2}}}\left(-\frac{3}{4}I_\beta\left(\mu(m^2)\right) + 2I_\gamma\left(\mu(m^2)\right)\right)\right). \qquad (86)$$

Note that $\lambda$ depends on the microscopic parameters of the model only through the single parameter $u = O(1/M)$. Now we can compute the integrals $I_\beta$ and $I_\gamma$ in the limit $m^2 \to 0$, which are convergent near $D = 6$:

$$\lim_{m^2 \to 0} I_\beta\left(\mu(m^2)\right) = \frac{1}{6}\Gamma\left(3 - \frac{D}{2}\right), \qquad (87)$$

$$\lim_{m^2 \to 0} I_\gamma\left(\mu(m^2)\right) = \frac{1}{2}\Gamma\left(3 - \frac{D}{2}\right). \qquad (88)$$

Thus in the limit $m^2 \to 0$

$$\lambda = u - \frac{7}{4}\frac{u^2}{(4\pi)^{\frac{D}{2}}}\Gamma\left(3 - \frac{D}{2}\right), \qquad (89)$$

from which

$$u \simeq \lambda + \frac{7}{4}\frac{\lambda^2}{(4\pi)^{\frac{D}{2}}}\Gamma\left(3 - \frac{D}{2}\right). \qquad (90)$$

Now, following the standard procedure (see Ref. [20], Chap. 8), we define the function $b(\lambda)$ as:

$$b(\lambda) \equiv m^2 \frac{\partial}{\partial m^2}\bigg|_{g \text{ fixed}} \lambda = \frac{1}{2}(D-6)u\frac{\partial}{\partial u}\bigg|_{m^2 \text{ fixed}} \lambda = \frac{1}{2}(D-6)\left(u - \frac{7}{2}\frac{u^2}{(4\pi)^{\frac{D}{2}}}\Gamma\left(3-\frac{D}{2}\right)\right). \quad (91)$$

From Eq. (90) we obtain:

$$b(\lambda) = \frac{1}{2}(D-6)\left(\lambda - \frac{7}{4}\frac{\lambda^2}{(4\pi)^{\frac{D}{2}}}\Gamma\left(3-\frac{D}{2}\right)\right). \quad (92)$$

We constructed $\lambda$ to be a dimensionless quantity that does not diverge at the critical point. For this reason, we can identify the critical value of $\lambda$ as the point at which the function $b(\lambda)$ is zero, as we discussed in Sec. 2. While a trivial zero is always present at $\lambda = 0$, for $D < 6$ we see that there also exists a non-trivial zero:

$$\lambda_c = \frac{4}{7}\frac{(4\pi)^{\frac{D}{2}}}{\Gamma\left(3-\frac{D}{2}\right)}. \quad (93)$$

As already remarked in Sec. 2, the value of $\lambda_c$ is universal, in the sense that it is no more dependent on the specific value of $g$, and thus on the specific problem we are considering, bond or site percolation. From this point the computation is really the same for the two cases, realizing universality between these two versions of the percolation problem.

Remembering that $m^2 \sim (\mu - \mu_c)^{2\nu}$, following standard computations [20], we define:

$$z(\lambda) \equiv \frac{\partial \mu}{\partial m^2} \sim m^{2D_1}, \quad (94)$$

where $D_1 = \frac{1}{2\nu} - 1$. We can thus compute it as:

$$D_1(\lambda) \equiv m^2 \frac{\partial}{\partial m^2}\bigg|_{g \text{ fixed}} \ln\big(z(\lambda)\big). \quad (95)$$

In the same way, for the computation of $\eta$ we need to define:

$$D_2(\lambda) \equiv \frac{\partial \ln \chi_2}{\partial \ln m^2}\bigg|_{g \text{ fixed}}, \quad \chi_2 \sim m^{2\frac{\eta-2}{2}}, \quad D_2(\lambda_c) = -1 + \frac{\eta}{2}. \quad (96)$$

We start from the computation of $z$:

$$z(\lambda) = a_l^2 \widehat{C}^{-\frac{2}{D}}\left(1 - \frac{1}{2}\frac{u}{(4\pi)^{\frac{D}{2}}}\frac{D-4}{2}I_\beta\big(\mu(m^2)\big) - \frac{1}{2}\frac{g}{(4\pi)^{\frac{D}{2}}}\frac{\partial}{\partial m^2}\Big(m^{D-4}I_\alpha\big(\mu(m^2)\big)\Big)\right), \quad (97)$$

where

$$\frac{\partial}{\partial m^2}\Big(m^{D-4}I_\alpha\big(\mu(m^2)\big)\Big) = -m^{D-6}\int_{\mu(m^2)/\Lambda^2}^{\infty}d\widehat{L}_a d\widehat{L}_b \frac{e^{-\widehat{L}_a-\widehat{L}_b}}{(\widehat{L}_a+\widehat{L}_b)^{\frac{D}{2}-1}} \equiv -m^{D-6}I_\alpha'\big(\mu(m^2)\big). \quad (98)$$

We can compute $I_\alpha'$:

$$\lim_{m^2 \to 0} I_\alpha'\big(\mu(m^2)\big) = \Gamma\left(3-\frac{D}{2}\right), \quad (99)$$

obtaining

$$z(\lambda) \propto 1 - \frac{u}{2}\frac{1}{(4\pi)^{\frac{D}{2}}}\Gamma\left(3-\frac{D}{2}\right)\frac{D-16}{12}, \quad (100)$$

and, from the definition of $D_1(\lambda)$, we arrive at the critical exponent $\nu$ in $D$ dimensions:

$$\nu_D = \frac{42}{84 + (6 - D)(D - 16)}. \tag{101}$$

The next exponent, $\eta$, requires the computation of $D_2(\lambda)$

$$D_2(\lambda) \equiv \left. \frac{\partial \ln \chi_2}{\partial \ln m^2} \right|_{g \text{ fixed}} = \left. \frac{m^2}{\chi_2} \frac{\partial \chi_2}{\partial m^2} \right|_{g \text{ fixed}} = -1 + \frac{\lambda}{2} \frac{1}{(4\pi)^{\frac{D}{2}}} I_\beta\left(\mu(m^2)\right)\left(\frac{D}{2} - 3\right), \tag{102}$$

which can be obtained using

$$\left. \frac{\partial \chi_2}{\partial m^2} \right|_{g \text{ fixed}} \propto -m^{-4} + \frac{1}{2} \frac{u}{(4\pi)^{\frac{D}{2}}} m^{-4} I_\beta\left(\mu(m^2)\right) \frac{D-8}{2}$$

$$= -m^{-4}\left(1 - \frac{1}{2} \frac{u}{(4\pi^{\frac{D}{2}})} I_\beta\left(\mu(m^2)\right) \frac{D-8}{2}\right), \tag{103}$$

$$\chi_2 \propto m^{-2}\left(1 + \frac{1}{2} \frac{u}{(4\pi^{\frac{D}{2}})} I_\beta\left(\mu(m^2)\right)\right), \tag{104}$$

from which we have

$$\eta_D = \frac{D-6}{21}. \tag{105}$$

**$\epsilon$-expansion** Given the results of Eqs. (101) and (105) in fixed dimension we can perform the computation in $D = 6 - \epsilon$:

$$\nu = \frac{1}{2} + \frac{5}{84}\epsilon + \mathcal{O}(\epsilon^2), \tag{106}$$

$$\eta = -\frac{1}{21}\epsilon + \mathcal{O}(\epsilon^2). \tag{107}$$

These results are, to first order in $\epsilon$, equal to the expansion of the standard field theory associated with the percolation problem [3, 6, 8–11].

# 7 Conclusion

In this article we have shown how to recover, at one-loop level of approximation, the results of the $\epsilon$-expansion for the critical exponents of the percolation problem on a $D$-dimensional regular lattice, by means of a new method, the $M$-layer construction. To do so, we computed the observables of interest for the case of site percolation in the non-percolating phase –the two- and three-point correlation functions, i.e. the probability that two or three sites belong to the same cluster– in properly chosen graphs at the leading orders. We then computed the $\epsilon$-expansion for the critical exponents, recovering, at first order, the same values already obtained for bond percolation using the $n \to 0$ continuation of the field theory applied to the Potts model with $n + 1$ states. Moreover, we have shown that within the $M$-layer construction the bond percolation problem differs from site percolation only for non-universal constants, which directly implies the universality between site and bond percolation in any dimension $D$. The analysis presented here clearly illustrates that the $M$-layer construction effectively allows one to extract quantitative information on the critical behavior even for problems which are not defined by a Hamiltonian, such as percolation.

We explained for the first time how this method can be applied to a known problem in order to obtain the $\epsilon$-expansion of the critical exponents. Recent studies have used the $M$-layer construction to derive non-trivial insights into models whose critical behavior is not yet completely understood [14–18], or to show that for well-known problems the one-loop results align with those from standard field theory [13, 24]. In this paper, we push this approach a step forward by showing how, applying the standard theoretical recipes of the renormalization group, one can extract the series for the critical exponents. We believe that this investigation could be highly beneficial in guiding the computation of critical exponents for problems where the standard RG approach is inapplicable [18].

Regarding the specific problem of percolation, it would be interesting to extend the calculations made in this work to the percolating phase $p > p_c$. In this sense, the preliminary calculation of the Ginzburg criterion at the bare order (i.e. without loops) has already been done using the $M$-layer construction, obtaining the known upper critical dimension, $D_U = 6$ [27]. To proceed further and obtain the values of the critical exponents in the percolating phase, it is necessary to calculate the same observables as Ref. [27] with the corrections due to the one-loop structures. We leave this analysis to future work.

## A  Identification of the constants in the $M$-layer expansion

In this Section we generalize the computation of the main text for the two and three-point function, with the goal of identifying the least number of constants that describe the loop expansion in the $M$-layer framework. In particular we will perform the computations for both site and bond percolation problems, highlighting the differences between the two. Starting with the site percolation, we write all the contributions, in Fourier space, of a generic $V_e$-point correlation, computed on a generic topology, $\mathcal{G}$, with $I$ lines, $V_e$ external points, $V_i$ internal vertices, $N_{loop}$ number of loops:

$$\overline{\widehat{C}_{V_e}(\{k_j\})}\Big|_{\mathcal{G}} = \frac{(2D)^{V_e}}{S(\mathcal{G}) M^{N_{loop}+V_e-1}} \left(\frac{(2D)!}{(2D-3)!}\right)^{V_i} \left(\prod_{i=1}^{I}\int dL_i\right)(2\pi)^D \delta^D\left(\sum_{j=1}^{V_e} k_j\right)$$

$$\times a_l^{-DV_i}\left(\int \prod_{l=1}^{N_{loop}} \frac{d^D q_l}{(2\pi)^D}\right)\left(\prod_{i=1}^{I}\widehat{\mathcal{N}}_{L_i}(\{q_l\},\{k_j\})\right) p^{I-2V_i} f(\mathcal{C}_{V_e,lc}) p^{\sum_{i=1}^{I} L_i}, \quad \text{(A.1)}$$

with the same convention for the Fourier transform used in the main text, Eq. (49). Notice that $\widehat{\mathcal{N}}_L$ are functions of linear combinations $g_i(\{q_l\},\{k_j\})$ of internal ($\{q_l\}$ for $l = 1,\ldots,N_{loop}$) and external momenta ($\{k_j\}$ for $j = 1,\ldots,V_e$), that ensure momentum conservation at each vertex. The factor $p^{I-2V_i}$ and the function $f(\mathcal{C}_{V_e,lc})$ come from Eqs. (36), (37), (42), (43) and (44). The first is the eventual extra factor $p$, which is present only for $\mathcal{C}_{2,lc}(\mathcal{G}_1;L)$ and $\mathcal{C}_{3,lc}(\mathcal{G}_3;\vec{L}')$, as can be checked by substituting the corresponding values for $I$ and $V_i$ (notice that the specific expression, $p^{I-2V_i}$, is valid only for three-degree vertices, for $V_i$ $d$-degree vertices it is $p^{I-(d-1)V_i}$ and can be generalized if vertices of different degree are present). The same goes for the factor $(2D-3)!$, whose generalization for a $d$-degree vertex is $(2D-d)!$. The function $f(\mathcal{C}_{V_e,lc})$ assumes the following values:

$$f(\mathcal{C}_{2,lc}(\mathcal{G}_1;L)) = 1, \quad \text{(A.2)}$$

$$f(\mathcal{C}_{2,lc}(\mathcal{G}_2;\vec{L})) = -1, \quad \text{(A.3)}$$

$$f(\mathcal{C}_{3,lc}(\mathcal{G}_3;\vec{L}')) = 1, \quad \text{(A.4)}$$

$$f(\mathcal{C}_{3,lc}(\mathcal{G}_4;\vec{L}'')) = -2, \quad \text{(A.5)}$$

$$f(\mathcal{C}_{3,lc}(\mathcal{G}_5;\vec{L}''')) = -1. \quad \text{(A.6)}$$

Notice that the diagrams we computed in this work are of the form of Eq. (A.1). We believe that higher order diagrams (with three-degree vertices only) for a generic $V_e$-point function obey it as well, but this hypothesis is not necessary for the results described in this paper. In principle we should repeat all the steps done from Eq. (A.1) to Eq. (A.6) for the bond percolation problem. Generalizing the arguments given in Sec. 5, we notice that the only difference with respect to the site percolation problem is the factor $p^{I-V_i}$ in Eq. (A.1), which is not present for bond percolation.

Let us continue with site percolation. Using the asymptotic expression of the NBP in Fourier space, Eq. (52), together with the rescaling of momenta and lengths, in Eq. (59), we arrive at

$$
\overline{\widehat{C}_{V_e}(\{k_j\})}\bigg|_{\mathcal{G}} = \frac{(2D)^{V_e}}{S(\mathcal{G}) M^{N_{loop}-1+V_e}} \left(\frac{(2D)!}{(2D-3)!}\right)^{V_i} \mu^{-I} \left(\prod_{i=1}^{I} \int d\widehat{L}_i\right)
$$

$$
\times (2\pi)^D \delta^D\left(\sum_{j=1}^{V_e} k_j\right) a_l^{-DV_i} \left(\int \prod_{l=1}^{N_{loop}} \frac{d^D \widehat{q}_l}{(2\pi)^D}\right) p^{I-2V_i} f(\mathcal{C}_{V_e,lc})
$$

$$
\times \left(\frac{\mu(2D-2)}{a_l^2}\right)^{\frac{D}{2}(N_{loop}-1)} \left(\frac{\mu(2D-2)}{a_l^2}\right)^{\frac{D}{2}} \left(\frac{2D}{2D-1}\right)^I \prod_{i=1}^{I} e^{-g_i(\{\widehat{q}_l\},\{\widehat{k}_j\})^2 \widehat{L}_i - \widehat{L}_i} a_l^{ID}, \quad \text{(A.7)}
$$

where $\widehat{k}$ is a function of $k$ according to (59). Note that, as done in the main text, we did not rescale the external momenta inside the delta function.

Given the known relations for $V_i$, $V_e$, $I$ and $N_{loop}$ in a generic diagram with internal vertices of degree three:

$$
V_i = V_e + 2(N_{loop}-1), \quad \text{and} \quad I = 2V_e + 3(N_{loop}-1), \tag{A.8}
$$

in Eq. (A.7) we can identify the following topology-dependent term:

$$
\frac{1}{S(\mathcal{G})} \left(\prod_{i=1}^{I} \int d\widehat{L}_i\right) \left(\int \prod_{l=1}^{N_{loop}} \frac{d^D \widehat{q}_l}{(2\pi)^D}\right) f(\mathcal{C}_{V_e,lc}) \prod_{i=1}^{I} e^{-g_i(\{\widehat{q}_l\},\{\widehat{k}_j\})^2 \widehat{L}_i - \widehat{L}_i}, \tag{A.9}
$$

and the following three factors:

- a constant to the power $(N_{loop}-1)$:

$$
\frac{1}{M} \left(\frac{(2D)!}{(2D-3)!}\right)^2 p^{-1} (2D-2)^{\frac{D}{2}} \left(\frac{2D}{2D-1}\right)^3 \mu^{\frac{D}{2}-3} \equiv \widehat{A} \mu^{\frac{D}{2}-3}, \tag{A.10}
$$

- a constant to the power $V_e$:

$$
\frac{1}{M} 2D \left(\frac{(2D)!}{(2D-3)!}\right) \left(\frac{2D}{2D-1}\right)^2 a_l^D \mu^{-2} \equiv \widehat{B} a_l^D \mu^{-2}, \tag{A.11}
$$

- an overall factor:

$$
(2\pi)^D \delta^D\left(\sum_{j=1}^{V_e} k_j\right) \left(\frac{\mu(2D-2)}{a_l^2}\right)^{\frac{D}{2}} \equiv (2\pi)^D \delta^D\left(\sum_{j=1}^{V_e} k_j\right) \mu^{D/2} \widehat{C} a_l^{-D}, \tag{A.12}
$$

as defined in Eqs. (56), (57) and (58). Again we notice that the same expressions are obtained in the bond percolation case, the only different one is the definition of $\widehat{A}$, given in this case by (60).

With the expression given in Eq. (A.7) it is possible to easily identify the relevant constants to perform the expansion in inverse powers of $M$. Let us remark that these are all non-universal quantities, being the critical exponents independent from them.

# B  Connection with field theoretical expressions

In this Section we show how to write the expressions for $\overline{\widehat{C}_{2,lc}}(k,k')$ and $\overline{\widehat{C}_{3,lc}}(k_1,k_2,k_3)$, Eqs. (54) and (55), in terms of scalar propagators, as in the corresponding field theory. To do so, starting from the mentioned equations, we first perform the integrals over the lengths with lower and upper limits of integration respectively $\mu/\Lambda^2$ and $\infty$. Notice that we are interested in the critical behavior, that is for $\mu \to 0$, thus we can set the lower limit to 0, which amounts to neglecting higher orders in $\mu$. The results are

$$
\begin{aligned}
\overline{\widehat{C}_2}(k,k') = {} & \frac{\widehat{C}\widehat{B}^2 a_l^D}{\widehat{A}\mu} \frac{1}{\widehat{k}^2+1} (2\pi)^D \delta^D(k+k') \\
& \times \left( 1 - \frac{\widehat{A}\mu^{\frac{D}{2}-3}}{2(\widehat{k}^2+1)} \int \frac{d^D\widehat{q}}{(2\pi)^D} \frac{1}{1+(\widehat{k}-\widehat{q})^2} \frac{1}{1+\widehat{q}^2} \right) + \mathcal{O}\left(\frac{1}{M^3}\right),
\end{aligned}
\tag{B.1}
$$

$$
\begin{aligned}
\overline{\widehat{C}_3}(k_1,k_2,k_3) = {} & \frac{\widehat{C}\widehat{B}^3 a_l^{2D}}{\widehat{A}\mu^3} \frac{(2\pi)^D \delta^D(k_1+k_2+k_3)}{(\widehat{k}_1^2+1)(\widehat{k}_2^2+1)(\widehat{k}_3^2+1)} \\
& \times \left( 1 - 2\widehat{A}\mu^{\frac{D}{2}-3} \int \frac{d^D\widehat{q}}{(2\pi)^D} \frac{1}{1+(\widehat{k}_2+\widehat{k}_3+\widehat{q})^2} \frac{1}{1+(\widehat{k}_2+\widehat{q})^2} \frac{1}{1+\widehat{q}^2} \right. \\
& \left. - \frac{1}{2} \frac{\widehat{A}\mu^{\frac{D}{2}-3}}{(\widehat{k}_2+\widehat{k}_3)^2+1} \int \frac{d^D\widehat{q}}{(2\pi)^D} \frac{1}{1+(\widehat{k}_2+\widehat{q})^2} \frac{1}{1+\widehat{q}^2} + \text{perm.} \right) + \mathcal{O}\left(\frac{1}{M^4}\right).
\end{aligned}
\tag{B.2}
$$

Next we rescale the momenta and we define the bare mass and coupling, respectively $m_b$ and $g_b$, according to:

$$
\widetilde{k} \equiv \mu^{\frac{1}{2}} a_l^{\frac{2D}{D+2}} \widehat{A}^{\frac{1}{D+2}} \widehat{B}^{-\frac{2}{D+2}} \widehat{k},
\tag{B.3}
$$

$$
m_b^2 \equiv \mu\, a_l^{-\frac{4D}{D+2}} \widehat{A}^{\frac{2}{D+2}} \widehat{B}^{-\frac{4}{D+2}},
\tag{B.4}
$$

$$
g_b \equiv a_l^{D\frac{D-6}{D+2}} \widehat{A}^{\frac{4}{D+2}} \widehat{B}^{\frac{D-6}{D+2}} \widehat{C}^{-2+\frac{3}{D}+\frac{D}{4}},
\tag{B.5}
$$

and we obtain

$$
\begin{aligned}
\overline{\widehat{C}_2}(\widetilde{k},\widetilde{k}') = {} & (2\pi)^D \delta^D(k+k') \left( \frac{1}{\widetilde{k}^2+m_b^2} - \frac{1}{2} g_b^2 \frac{1}{(\widetilde{k}^2+m_b^2)^2} \int \frac{d^D\widetilde{q}}{(2\pi)^D} \frac{1}{(\widetilde{k}-\widetilde{q})^2+m_b^2} \frac{1}{\widetilde{q}^2+m_b^2} \right) \\
& + \mathcal{O}\left(\frac{1}{M^3}\right),
\end{aligned}
\tag{B.6}
$$

$$
\begin{aligned}
\overline{\widehat{C}_3}(k_1,k_2,k_3) = {} & \frac{1}{(\widetilde{k}_1^2+m_b^2)(\widetilde{k}_2^2+m_b^2)(\widetilde{k}_3^2+m_b^2)} (2\pi)^D \delta^D(\widetilde{k}_1+\widetilde{k}_2+\widetilde{k}_3) \\
& \times \left( g_b - 2g_b^3 \int \frac{d^D\widetilde{q}}{(2\pi)^D} \frac{1}{(\widetilde{k}_2+\widetilde{k}_3+q)^2+m_b^2} \frac{1}{(\widetilde{k}_2+\widetilde{q})^2+m_b^2} \frac{1}{\widetilde{q}^2+m_b^2} \right. \\
& \left. - \frac{1}{2} g_b^3 \frac{1}{(\widetilde{k}_2+\widetilde{k}_3)^2+m_b^2} \int \frac{d^D\widetilde{q}}{(2\pi)^D} \frac{1}{(\widetilde{k}_2+\widetilde{q})^2+m_b^2} \frac{1}{\widetilde{q}^2+m_b^2} + \text{perm.} \right) \\
& + \mathcal{O}\left(\frac{1}{M^4}\right),
\end{aligned}
\tag{B.7}
$$

which are the results of the corresponding field theory associated with the percolation problem [7,9].

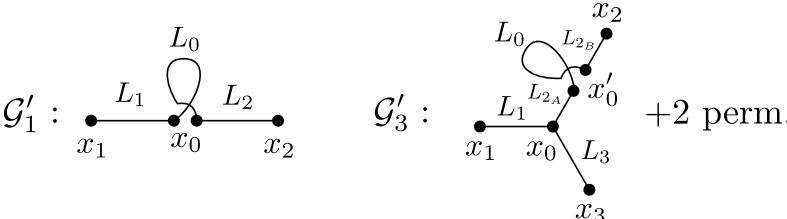

Figure 4: Less divergent diagrams that contribute to the two-point function. Notice that the two couple of vertices in $x_0$ and $x'_0$ belong to two different layers while on the projection they are superimposed. The two lines of length $L_0$ coil themselves in the $M$-layer lattice, in such a way that the projection on the original lattice looks like a loop.

As a last remark we notice that it is not always possible to write the results of the $M$-layer construction in terms of scalar propagators. For the percolation problem, the observables computed on a given topology, such as Eqs. (36) or (37), are powers of the probability $p$ to some combination of the lengths of the lines, thus the integrals over the lengths give the scalar propagator factors. For a generic problem the expressions of the observables can be more complicated functions of the lengths (see Refs. [14,17] as an example) and the corresponding integrals do not give the simple structure of a scalar propagator. On the other hand, for simple problems, whose field theoretical analysis is clear, the propagator structure is recovered by means of the $M$-layer construction [24].

It is also interesting to note that the integrals occurring in field theories are *actually computed* through the application of formulas like the following:

$$\frac{1}{k^2+m^2} = \int_0^\infty e^{-l(k^2+m^2)}dl\,, \tag{B.8}$$

see *e.g.* the appendix to Chap. 5 in [20]. This amounts to go from Eqs. (B.1) and (B.2) back to Eqs. (54) and (55). Thus the $M$-layer approach directly gives expressions in the above treatable form. Furthermore, the integration variable $l$, that seems artificial in field theory, has instead the natural meaning of the length of the internal lines of the diagrams in the $M$-layer approach.

## C   Other diagrams

In this appendix we take into account other possible diagrams of order $\mathcal{O}(1/M^2)$ that may contribute to the two-point correlation. As discussed in Ref. [24], the computation of the line without loop should be corrected to $\mathcal{O}(1/M^2)$ by diagram $\mathcal{G}'_1$ in Fig. 4, with the corresponding weight: $W(\mathcal{G}'_1) = 1/M(1-1/M)$. While the contribution of $\mathcal{G}'_1$ at order $\mathcal{O}(1/M)$ is already included in Eq. (40), its contribution at order $\mathcal{O}(1/M^2)$ is not included there because $\mathcal{G}'_1$ diverges with a lower power of $\mu$ with respect to $\mathcal{G}_2$, which also contributes at order $\mathcal{O}(1/M^2)$.

The contribution of $\mathcal{G}'_1$ at order $\mathcal{O}(1/M^2)$ is

$$-\frac{(2D)^2}{M^2}\frac{(2D)!}{(2D-4)!}\sum_{L_1,L_0,L_2}\sum_{x_0}\mathcal{N}_{L_1}(x_1,x_0)\mathcal{N}_{L_0}(x_0,x_0)\mathcal{N}_{L_2}(x_0,x_2)\mathcal{C}_{2,lc}(\mathcal{G}'_1;L_1,L_0,L_2), \tag{C.1}$$

where

$$\mathcal{C}_{2,lc}(\mathcal{G}'_1;L_1,L_0,L_2) = \mathcal{C}_{2,lc}(\mathcal{G}_1;L_1+L_0+L_2) = p\,p^{L_1+L_0+L_2}\,. \tag{C.2}$$

In Fourier space, using Eqs. (51) and (52), it becomes:

$$
-(2\pi)^D \delta^D(k_1+k_2)\frac{(2D)^2}{M^2}\frac{(2D)!}{(2D-4)!}\left(\frac{2D}{2D-1}\right)^3 a_l^{2D}\, p\, \frac{1}{\frac{a_l^2}{2D-2}k_1^2+\mu}\frac{1}{\frac{a_l^2}{2D-2}k_2^2+\mu}
$$
$$
\times \int \frac{d^D k_0}{(2\pi)^D}\int_{\mu/\Lambda^2}^{\infty} d\widehat{L}_0 e^{-\left(\frac{a_l^2}{\mu(2D-2)}k_0^2+1\right)\widehat{L}_0}, \quad \text{(C.3)}
$$

which can be rewritten by scaling all the momenta, $\widehat{k}\equiv k\,\frac{a_l}{\sqrt{\mu(2D-2)}}$, apart from the ones in the delta function, as:

$$
-(2\pi)^D \delta^D(k_1+k_2)\frac{(2D)^2}{M^2}\frac{(2D)!}{(2D-4)!}\left(\frac{2D}{2D-1}\right)^3 \mu^{\frac{D}{2}-3}\frac{a_l^D\, p(2D-2)^{\frac{D}{2}}}{(\widehat{k}_1^2+1)(\widehat{k}_2^2+1)}\int \frac{d^D\widehat{k}_0}{(2\pi)^D}\frac{1}{\widehat{k}_0^2+1}\propto \frac{\mu^{\frac{D}{2}-3}}{M^2},
$$
$$
\text{(C.4)}
$$

where, as usual, we neglected higher orders in $\mu$ setting the lower limit of the length integration to 0. The other contribution to order $\mathcal{O}(1/M^2)$ is from diagram $\mathcal{G}_2$, repeating the same steps we have

$$
-(2\pi)^D \delta^D(k_1+k_2)\frac{(2D)^2}{2M^2}\left(\frac{(2D)!}{(2D-3)!}\right)^2\left(\frac{2D}{2D-1}\right)^4 \mu^{\frac{D}{2}-4}\frac{a_l^D(2D-2)^{\frac{D}{2}}}{(\widehat{k}_1^2+1)(\widehat{k}_2^2+1)}
$$
$$
\times \int \frac{d^D\widehat{k}}{(2\pi)^D}\frac{1}{\widehat{k}^2+1}\frac{1}{(\widehat{k}_1-\widehat{k})^2+1}\propto \frac{\mu^{\frac{D}{2}-4}}{M^2}, \quad \text{(C.5)}
$$

from which it is clear that near the critical point, $\mu \sim 0$, the contribution of $\mathcal{G}_1'$ can be neglected with respect to the one of $\mathcal{G}_2$. Analogously, diagram $\mathcal{G}_3'$, is negligible with respect to $\mathcal{G}_4$ and $\mathcal{G}_5$. Thus the computations for the three-point correlation function of the main text give the correct critical behavior.

It is also possible to generalize this argument, at least in the case of the percolation problem. Since for each line of the diagram a factor proportional to $\mu^{-1}(\widehat{k}^2+1)^{-1}$ appears, we understand that, at a given order in $\mathcal{O}(1/M)$ the most divergent diagrams, in the limit $\mu \to 0$ are the ones with the largest number of lines. This argument is not valid generally for any problem or model. Indeed, the computation of the observables on a given diagram is the only model-dependent part of the $M$-layer procedure and in general the result can be a non-trivial function of the lengths, as we noticed at the end of appendix B.

## D  Four-point correlation function

We present, in this appendix, the computation for the most divergent contributions to the four-point correlation function in the site percolation problem, the same result can be obtained for the bond percolation with the arguments given in Sec. 5. All the possible topologies, with only three and four-degree vertices, are shown in Fig. 5. Along the lines of the reasoning given for neglecting $\mathcal{G}_1'$ with respect to $\mathcal{G}_2$ we identify the most divergent diagrams to each $\mathcal{O}(1/M)$ order for the four-point correlation function simply considering the diagrams with the largest number of lines. It turns out that the relevant diagrams, for the four-point function, are the ones shown in Fig. 6: $\mathcal{G}_7$ to order $\mathcal{O}(1/M^3)$, $\mathcal{G}_9$, $\mathcal{G}_{12}$ and $\mathcal{G}_{13}$ to order $\mathcal{O}(1/M^4)$. Notice that, in principle, we should have considered also diagrams with vertices of degree larger than four, but they all have, at one loop order, fewer lines than the ones we included in Fig. 6, thus they are less divergent near the critical point $\mu \sim 0$. Now we can write the contributions of

Figure 5: Diagrams contributing to the four-point correlation function up to one loop.

the identified diagrams:

$$\overline{C_4(x_1, x_2, x_3, x_4)} = \frac{1}{M^3} \sum_{\vec{L}} \sum_{x_0, x_0'} \mathcal{N}(\mathcal{G}_7; \vec{L}; x_1, x_2, x_3, x_4, x_0, x_0') \mathcal{C}_{4,lc}(\mathcal{G}_7; \vec{L})$$

$$+ \frac{1}{M^4} \sum_{\vec{L}'} \sum_{\{x_i'\}, i=1,...,4} \mathcal{N}(\mathcal{G}_9; \vec{L}'; x_1, x_2, x_3, x_4, x_1', x_2', x_3', x_4') \mathcal{C}_{4,lc}(\mathcal{G}_9; \vec{L}')$$

$$+ \frac{1}{M^4} \sum_{\vec{L}'} \sum_{x_1', x_0, x_0', x_0''} \mathcal{N}(\mathcal{G}_{10}; \vec{L}'; x_1, x_2, x_3, x_4, x_1', x_0, x_0', x_0'') \mathcal{C}_{4,lc}(\mathcal{G}_{10}; \vec{L}')$$

$$+ \frac{1}{2M^4} \sum_{\vec{L}'} \sum_{\{x_i'\}, i=1,...,4} \mathcal{N}(\mathcal{G}_{12}; \vec{L}'; x_1, x_2, x_3, x_4, x_1', x_2', x_3', x_4') \mathcal{C}_{4,lc}(\mathcal{G}_{12}; \vec{L}')$$

$$+ \frac{1}{2M^4} \sum_{\vec{L}''} \sum_{x_1', x_0, x_0', x_0''} \mathcal{N}(\mathcal{G}_{13}; \vec{L}''; x_1, x_2, x_3, x_4, x_1', x_0, x_0', x_0'') \mathcal{C}_{4,lc}(\mathcal{G}_{13}; \vec{L}'') + \mathcal{O}\left(\frac{1}{M^5}\right), \quad \text{(D.1)}$$

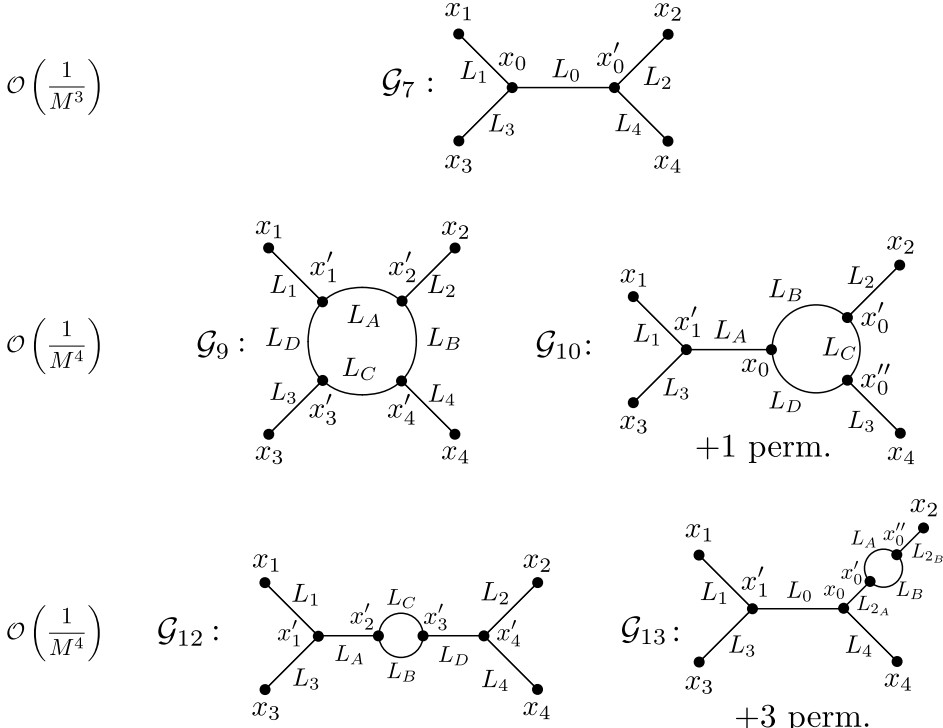

Figure 6: Most divergent diagrams contributing to the four-point correlation functions up to one loop near the critical point.

where the lengths are defined as $\vec{L} = (L_0, L_1, L_2, L_3, L_4)$, $\vec{L}' = (L_1, L_2, L_3, L_4, L_A, L_B, L_C, L_D)$, $\vec{L}'' = (L_1, L_3, L_4, L_{2_A}, L_{2_B}, L_A, L_B, L_C, L_D)$, and the NBPs:

$$\mathcal{N}(\mathcal{G}_7; \vec{L}; x_1, x_2, x_3, x_4, x_0, x_0') \tag{D.2}$$
$$= (2D)^4 \left( \frac{(2D)!}{(2D-3)!} \right)^2 \prod_{i=1,3} \mathcal{N}_{L_i}(x_i, x_0) \prod_{i=2,4} \mathcal{N}_{L_i}(x_i, x_0) \mathcal{N}_{L_0}(x_0, x_0'),$$

$$\mathcal{N}(\mathcal{G}_9; \vec{L}'; x_1, x_2, x_3, x_4, x_1', x_2', x_3', x_4') \tag{D.3}$$
$$= (2D)^4 \left( \frac{(2D)!}{(2D-3)!} \right)^4 \prod_{i=1}^{4} \mathcal{N}_{L_i}(x_i, x_i') \mathcal{N}_{L_A}(x_1', x_2') \mathcal{N}_{L_B}(x_2', x_4') \mathcal{N}_{L_C}(x_3', x_4') \mathcal{N}_{L_D}(x_3', x_1'),$$

$$\mathcal{N}(\mathcal{G}_{10}; \vec{L}'; x_1, x_2, x_3, x_4, x_1', x_0, x_0', x_0'') \tag{D.4}$$
$$= (2D)^4 \left( \frac{(2D)!}{(2D-3)!} \right)^4 \prod_{i=1}^{4} \mathcal{N}_{L_i}(x_i, x_i') \mathcal{N}_{L_A}(x_1', x_0) \mathcal{N}_{L_B}(x_0, x_0') \mathcal{N}_{L_C}(x_0', x_0'') \mathcal{N}_{L_D}(x_0, x_0''),$$

$$\mathcal{N}(\mathcal{G}_{12}; \vec{L}'; x_1, x_2, x_3, x_4, x_1', x_2', x_3', x_4') = (2D)^4 \left( \frac{(2D)!}{(2D-3)!} \right)^4 \mathcal{N}_{L_1}(x_1, x_1') \tag{D.5}$$
$$\times \mathcal{N}_{L_2}(x_2, x_4') \mathcal{N}_{L_3}(x_3, x_1') \mathcal{N}_{L_4}(x_4, x_4') \mathcal{N}_{L_A}(x_1', x_2') \mathcal{N}_{L_B}(x_2', x_3') \mathcal{N}_{L_C}(x_2', x_3') \mathcal{N}_{L_D}(x_4', x_3'),$$

$$\mathcal{N}(\mathcal{G}_{13}; \vec{L}''; x_1, x_2, x_3, x_4, x_1', x_0, x_0', x_0'') = (2D)^4 \left( \frac{(2D)!}{(2D-3)!} \right)^4 \mathcal{N}_{L_1}(x_1, x_1') \tag{D.6}$$
$$\times \mathcal{N}_{L_3}(x_3, x_1') \mathcal{N}_{L_4}(x_4, x_0) \mathcal{N}_{L_0}(x_1', x_0) \mathcal{N}_{L_{2_B}}(x_2, x_0'') \mathcal{N}_{L_{2_A}}(x_0, x_0') \mathcal{N}_{L_A}(x_0', x_0'') \mathcal{N}_{L_B}(x_0', x_0''),$$

and finally the observables

$$\mathcal{C}_{4,lc}(\mathcal{G}_7;\vec{L}) = p\, p^{L_1+L_2+L_3+L_4+L_0}\,, \tag{D.7}$$

$$\mathcal{C}_{4,lc}(\mathcal{G}_9;\vec{L}') = -3p^{L_1+L_2+L_3+L_4+L_A+L_B+L_C+L_D}\,, \tag{D.8}$$

$$\mathcal{C}_{4,lc}(\mathcal{G}_{10};\vec{L}') = -2p^{L_1+L_2+L_3+L_4+L_A+L_B+L_C+L_D}\,, \tag{D.9}$$

$$\mathcal{C}_{4,lc}(\mathcal{G}_{12};\vec{L}') = -p^{L_1+L_2+L_3+L_4+L_A+L_B+L_C+L_D}\,, \tag{D.10}$$

$$\mathcal{C}_{4,lc}(\mathcal{G}_{13};\vec{L}'') = -p^{L_1+L_3+L_4+L_0+L_{2_A}+L_{2_B}+L_A+L_B}\,. \tag{D.11}$$

Since the identified diagrams, $\mathcal{G}_7$, $\mathcal{G}_9$, $\mathcal{G}_{10}$, $\mathcal{G}_{12}$ and $\mathcal{G}_{13}$, contain only three-degree vertices, we can use the generic equation derived in App. A for this kind of vertices, Eq. (A.7), where

$$f\big(\mathcal{C}_{4,lc}(\mathcal{G}_7;\vec{L}',L_4,L_0)\big) = 1\,, \tag{D.12}$$

$$f\big(\mathcal{C}_{4,lc}(\mathcal{G}_9;\vec{L})\big) = -3\,, \tag{D.13}$$

$$f\big(\mathcal{C}_{4,lc}(\mathcal{G}_{10};\vec{L})\big) = -2\,, \tag{D.14}$$

$$f\big(\mathcal{C}_{4,lc}(\mathcal{G}_{12};\vec{L}'')\big) = -1 = f\big(\mathcal{C}_{4,lc}(\mathcal{G}_{13};\vec{L}''')\big)\,, \tag{D.15}$$

and $S(\mathcal{G}_7) = S(\mathcal{G}_9) = S(\mathcal{G}_{10}) = 1$, $S(\mathcal{G}_{12}) = 2 = S(\mathcal{G}_{13})$:

$$
\begin{aligned}
\overline{\widehat{C}_{4,lc}}(\{k_i\}_{i=1,\dots,4}) = {}& (2\pi)^D \delta^D\left(\sum_{i=1}^4 k_i\right) \frac{\widehat{B}^4\, a_l^{3D}\, \widehat{C}}{\widehat{A}\mu^5} \prod_{i=1}^4 \frac{1}{\widehat{k}_i^2+1}\Bigg( \int d\widehat{L}_0\, e^{-\widehat{L}_0-(\widehat{k}_1+\widehat{k}_3)^2\widehat{L}_0} \\
& -3\widehat{A}\mu^{\frac{D}{2}-3} \prod_{i=A,B,C,D}\int d\widehat{L}_i\, e^{-\widehat{L}_i} \int \frac{d^D\widehat{q}}{(2\pi)^D} e^{-\widehat{q}^2\widehat{L}_A-(\widehat{q}+\widehat{k}_2)^2\widehat{L}_B-(\widehat{q}+\widehat{k}_2+\widehat{k}_3)^2\widehat{L}_C-(\widehat{k}_1-\widehat{q})^2\widehat{L}_D} \\
& -2\widehat{A}\mu^{\frac{D}{2}-3} \prod_{i=A,B,C,D}\int d\widehat{L}_i\, e^{-\widehat{L}_i} e^{-(\widehat{k}_1+\widehat{k}_3)^2\widehat{L}_A} \int \frac{d^D\widehat{q}}{(2\pi)^D} e^{-\widehat{q}^2\widehat{L}_B-(\widehat{q}+\widehat{k}_2)^2\widehat{L}_C-(\widehat{q}+\widehat{k}_2+\widehat{k}_4)^2\widehat{L}_D} \\
& -2\widehat{A}\mu^{\frac{D}{2}-3} \prod_{i=A,B,C,D}\int d\widehat{L}_i\, e^{-\widehat{L}_i} e^{-(\widehat{k}_2+\widehat{k}_4)^2\widehat{L}_A} \int \frac{d^D\widehat{q}}{(2\pi)^D} e^{-\widehat{q}^2\widehat{L}_B-(\widehat{q}-\widehat{k}_1)^2\widehat{L}_C-(\widehat{q}-\widehat{k}_1-\widehat{k}_3)^2\widehat{L}_D} \\
& -\frac{\widehat{A}\mu^{\frac{D}{2}-3}}{2} \prod_{i=A,B,C,D}\int d\widehat{L}_i\, e^{-\widehat{L}_i} e^{-(\widehat{k}_1+\widehat{k}_3)^2(\widehat{L}_A+\widehat{L}_D)} \int \frac{d^D\widehat{q}}{(2\pi)^D} e^{-\widehat{q}^2\widehat{L}_B-(\widehat{q}-\widehat{k}_1-\widehat{k}_3)^2\widehat{L}_C} \\
& -\frac{\widehat{A}\mu^{\frac{D}{2}-3}}{2} \sum_{j=1}^4 \prod_{i=0,A,B,j_A,j_B}\int d\widehat{L}_i\, e^{-\widehat{L}_i} e^{-(\widehat{k}_1+\widehat{k}_3)^2\widehat{L}_0-\widehat{k}_2^2(\widehat{L}_{j_A}+L_{j_B})} \int \frac{d^D\widehat{q}}{(2\pi)^D} e^{-\widehat{q}^2\widehat{L}_A-(\widehat{q}+\widehat{k}_j)^2\widehat{L}_B} \Bigg) \\
& +\mathcal{O}\left(\frac{1}{M^5}\right)\,. \tag{D.16}
\end{aligned}
$$

As for the two and three-point correlation functions, we define $\chi_4(\mu)$ as the four-point correlation function at zero external momenta, divided by $a_l^{3D}$ and without the factor $(2\pi)^D\delta^D\left(\sum_{i=1}^4 k_i\right)$:

$$\chi_4(\mu) = \frac{\widehat{B}^4\widehat{C}}{\widehat{A}\mu^5}\left(1 - \frac{5}{2}\frac{\widehat{A}\mu^{\frac{D}{2}-3}}{(4\pi)^{\frac{D}{2}}}I_\alpha(\mu) - 4\frac{\widehat{A}\mu^{\frac{D}{2}-3}}{(4\pi)^{\frac{D}{2}}}I_\gamma(\mu) - 3\frac{\widehat{A}\mu^{\frac{D}{2}-3}}{(4\pi)^{\frac{D}{2}}}I_\delta(\mu)\right)\,, \tag{D.17}$$

where $I_\alpha$ and $I_\gamma$ are defined in Eqs. (76) and (82) respectively, while

$$I_\delta(\mu) \equiv \int_{\frac{\mu}{\Lambda^2}}^\infty d\widehat{L}_A d\widehat{L}_B d\widehat{L}_C d\widehat{L}_D \frac{e^{-\widehat{L}_A-\widehat{L}_B-\widehat{L}_C-\widehat{L}_D}}{\left(\widehat{L}_A+\widehat{L}_B+\widehat{L}_C+\widehat{L}_D\right)^{\frac{D}{2}}}\,, \tag{D.18}$$

and consequently

$$\lim_{\mu\to 0} I_\delta(\mu) = \frac{6-D}{12}\Gamma\left(3-\frac{D}{2}\right)\,. \tag{D.19}$$

Using the relation between $\mu$ and $m^2$, Eq. (79), we can write $\chi_4$ as a function of $m^2$

$$\chi_4\left(\mu(m^2)\right) = \frac{\widehat{B}^4 \widehat{C}^{\frac{10}{D}+1}}{\widehat{A} a_l^{10}} m^{-10} \left(1 + \frac{5}{2} \frac{u}{(4\pi)^{\frac{D}{2}}} I_\beta\left(\mu(m^2)\right)\right.$$
$$\left. - 4\frac{u}{(4\pi)^{\frac{D}{2}}} I_\gamma\left(\mu(m^2)\right) - 3\frac{u}{(4\pi)^{\frac{D}{2}}} I_\delta\left(\mu(m^2)\right)\right), \quad \text{(D.20)}$$

where $u$ is the bare coupling constant, defined in Eq. (85). Now we can look at the scaling, near the critical point $m^2 \to 0$, of the four-point function

$$D_4(\lambda) \equiv \left.\frac{\partial \ln \chi_4\left(\mu(m^2 \simeq 0)\right)}{\partial \ln m^2}\right|_{g \text{ fixed}}, \qquad \chi_4\left(\mu(m^2 \simeq 0)\right) \sim m^{2D_4(\lambda_c)}. \quad \text{(D.21)}$$

Using the expression of $\chi_4(m^2)$ we have

$$D_4(\lambda_c) = \frac{1}{42}\Big(D(3D-55)+12\Big). \quad \text{(D.22)}$$

We can now compare with the scaling of the four-point function

$$\chi_q\left(\mu(m^2 \simeq 0)\right) \sim m^{2D_q(\lambda_c)}, \quad D_q(\lambda_c) = \frac{q}{4}\eta - \frac{q}{2} + \frac{D}{2}\left(1 - \frac{q}{2}\right), \quad \text{(D.23)}$$

with $q = 4$, which gives the expected result of Eq. (105)

$$\eta_D = \frac{D-6}{21}. \quad \text{(D.24)}$$

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
