# Peer review of "Bethe $M$-layer construction for the percolation problem"

_SciPost Physics, doi:SciPost Phys. 18, 030 (2025)_

## Round 1 · Referee Report · Anonymous (Referee 1) · 2024-11-9

Strengths

  1. The paper explores the capability of a relatively recently introduced new expansion around Bethe lattice solution in Statistical Mechanics
  2. It shows that it is able to reproduce old well established results of percolation theory without need of the 1-state Potts trick.
  3. Quite well written and clear.

Weaknesses

  1. The paper is very technical.
  2. Not well separated the result of the new expansion from standard RG reasoning.

Report

This paper is part of an ambitious program of constructing a theory of strongly fluctuating disordered systems in finite D, starting from an expansion around the Bethe-Peierls limit of models on Random Graphs (Bethe Lattice). Despite the increase in difficulty with respect to the usual field theoretical expansion around homogeneous saddle points, this approach could give a better starting point in the perturbation theory for finite D systems in cases where the Bethe Lattice and the limit of full connectivity give different phase diagrams. The percolation problem is an example of this kind of problem, being well defined (and a classical problem) on random graphs, and not defined in a fully connected graph. As a result, in the approach of the authors the Fortuin-Kasteleyn trick of the 1-state Potts model (although possible) is not necessary to map the problem in a field theory. The authors therefore study the percolation on an M-layer graph, which is built to interpolate between the D-dimensional hypercubic lattice for M=1, and the Random Regular graph with degree 2D for large M.  The method they use is a 1/M expansion that they can convert in an epsilon expansion around dimension 6 for percolation. Given the difficulty of the approach the paper is limited to the order 1/M (giving the first non trivial order in epsilon) which is coherent with the usual Potts model approach to the same order. While not giving new results on the percolation problem -except perhaps the explicit verification that site and bond percolations have the same exponents-, this paper provides a proof of concept that expanding around the Bethe lattice gives rise to the field theoretical description of the ordinary RG loop expansion. In order to study the transition, the authors consider a rescaled a-dimensional ratio of  second and third moment of the cluster size distribution, identified as physical coupling constant of the theory, and use 1/M perturbation theory to analyse it according to usual RG when the critical point is approached. The diagrammatic rules of the theory, which were defined in previous papers of the same collaboration, are discussed at an acceptable level of detail. Probably, the main result of the paper is the equivalence, stated in lines 368-371 and shown in appendix B, of the two and three body correlations in Fourier space with the ones of the ordinary theory. Once this result is obtained, the computation of the beta function and the exponents at the order $\epsilon$ is textbook RG theory. 
I think that illustrating that a new promising method reproduces known results in simple cases is interesting and the paper is worth publishing. Despite the highly technical content I found the paper readable and quite well written. I found the description of the steps required in the computation of the correlation functions and its repetition in the case of the two body and the three body functions quite clear and useful.

Requested changes

The presentation of the paper could be improved by separating more clearly the part specific to the M-layer expansion from the part  which is standard RG theory. Also, since the difference between site and bond percolation is very small, I think that the author could discuss the two problems in parallel in the same section. 

Side remarks
The remark on lines 106-107 after eq. 11 could result confusing: It could be useful to state that eq. 11 is the result of the 1/M expansion, while the fact that $u$ diverges and $\lambda$ remains finite are exact results.

 It could be useful to show in the main body how $M$ disappears from the computation and one has universality for different values of M as announced in the introduction. 

In the discussion on percolation on the Bethe lattice,  references and a small explanation of the derivation of eq. 19 and 20 should be given.  

The terms 'topological loops' and 'topological diagrams' should be explained: please give the definition.  

Same comment for 'line connected observables' for which the reader is referred to [12]. I think that a small clarification would make the reader's life easier.

Recommendation

Ask for minor revision

  • validity: high
  • significance: good
  • originality: good
  • clarity: high
  • formatting: good
  • grammar: good

Author:  Saverio Palazzi  on 2024-12-02  [id 5014]

(in reply to Report 1 on 2024-11-09)
Category:
remark
answer to question
correction

The authors would like to thank the referee for the thoughtful report of our manuscript. In particular we address each of the points raised and provide clarifications and revisions:

Referee: "The presentation of the paper could be improved by separating more clearly the part specific to the M-layer expansion from the part which is standard RG theory. Also, since the difference between site and bond percolation is very small, I think that the author could discuss the two problems in parallel in the same section. "
Authors: As suggested we divided Section 5 in two Sections. The second one (Section 6 of the new version) is called “Computations of critical exponents”. This allows one to better distinguish the $M$-layer computations from the standard RG ones. Moreover we eliminated the old Section 6, about the bond percolation, and incorporated these results in Section 5, that now treats both the site and bond percolation problems in parallel.

Side remarks
Referee: "The remark on lines 106-107 after eq. 11 could result confusing: It could be useful to state that eq. 11 is the result of the 1/M expansion, while the fact that u diverges and λ remains finite are exact results."
Authors: In order to clarify that the divergence of $u$ and the convergence of $\lambda$ is not a consequence of the perturbative relation between $\lambda$ and $u$, given in Eq (11), we added a comment there.

Referee: "It could be useful to show in the main body how M disappears from the computation and one has universality for different values of M as announced in the introduction. "
Authors: After the equation for the critical value of $\lambda$ (Eq. (15) of the new version) we added a comment on the universality, in particular on the disappearance of any microscopic quantity.

Referee:"In the discussion on percolation on the Bethe lattice, references and a small explanation of the derivation of eq. 19 and 20 should be given. "
Authors: We integrated a short justification for Eqs. (19) and (20) (respectively Eq. (20) and Eq. (21) in the new version) with an explanatory figure, Fig.1 in the revised paper.

Referee:" The terms 'topological loops' and 'topological diagrams' should be explained: please give the definition. "
Authors: We added the missing definitions for "topological loop" and "topological diagram" at the beginning of Section 4 (lines 222 and 242 of the new version respectively).

Referee: "Same comment for 'line connected observables' for which the reader is referred to [12]. I think that a small clarification would make the reader's life easier."
Authors: We added an operative definition to compute "line-connected" observables in the "step 3" item of Section 4, in line 291 of the new version.

---

## Round 1 · Referee Report · Anonymous (Referee 2) · 2024-11-18

Strengths

1- Very well-written and nicely structured. 2- Detailed calculations useful for site and bond percolation problems as well as for a large class of disordered systems.

Weaknesses

1 - At times, the paper appears extremely technical.

Report

The paper “Bethe M-layer construction for the percolation problem” by Angelini, Palazzi et al. introduces an innovative approach to addressing both site and bond percolation problems using the M-layer expansion. In the large M-limit, the Bethe approximation, which typically fails for non-tree-like topologies, becomes asymptotically exact. The quantity 1/M serves a small perturbative parameter, enabling a systematic expansion around the Bethe solution.

The authors specifically show how this framework reproduces the critical exponents in perturbation theory —depending on the number of spatial loops considered — without relying on the $ n \rightarrow 0$ analytical continuation of the Potts model as originally proposed by Kasteleyn and Fortuin. Moreover, they verify the equivalence of the critical exponents for site and bond percolation problems within this approach.
This methodology is highly versatile and could be potentially generalized to other classical and quantum complex systems for which a finite-dimensional solution remains elusive. Its applicability extends also to problems for which standard renormalization group approaches are undoable.
I am strongly supportive of seeing the paper published in "SciPost Physics". I only have a few concerns on which I would appreciate further clarification from the authors. Below is a detailed list of suggested changes.

Requested changes

1- On page 4, line 104, the authors introduce the constant $u \equiv g m^{D-6}$ without first discussing the upper critical dimension of the model. It would be helpful to elaborate on how the upper critical dimension arises and to clarify its relevance to the choice of diagrams included in the following analysis.

2- Section 3 is rather cryptic for a reader who is not an expert in the field or familiar with cavity equations. Providing additional explanations would make the content more accessible to a broader audience.

3- On page 9, different "topological diagrams" are considered, both with three-degree vertices ($G_2$) and four-degree vertices ($G’$). These diagrams turn out to be relevant, respectively, in a $\phi^3$ and a $\phi^4$ theory. At the end of the calculation, only the diagrams $G_1$ and $G_2$ are retained, while $G’$ and the tadpoles are neglected (at least for $p<p_c$). Is this the reasoning used to argue that the upper critical dimension of the theory is $D^{u}=6$, whereby the surviving diagrams are exclusively those associated with a cubic field theory?

4- While the symmetry factors for the selected diagrams are immediate, it would be valuable to include a more detailed explanation of how symmetry factors are determined for more complex diagrams. I, therefore, encourage the authors to either include the calculation steps in an appendix or to refer to a suitable source. I have a similar concern about the “Bond percolation" section in which the differences/similarities are presented in an overly concise manner.

Recommendation

Publish (meets expectations and criteria for this Journal)

  • validity: high
  • significance: high
  • originality: good
  • clarity: high
  • formatting: good
  • grammar: excellent

Author:  Saverio Palazzi  on 2024-12-02  [id 5015]

(in reply to Report 2 on 2024-11-18)
Category:
remark
answer to question
correction

The authors would like to thank the referee for the thoughtful report of our manuscript. In particular we address each of the points raised in the report and provide clarifications and revisions:

Referee: "1- On page 4, line 104, the authors introduce the constant $u\equiv g m^{D−6}$ without first discussing the upper critical dimension of the model. It would be helpful to elaborate on how the upper critical dimension arises and to clarify its relevance to the choice of diagrams included in the following analysis."
Authors: In order to distinguish between results from the $M$-layer construction and results from standard $\epsilon$-expansion we divided Section 5 in two, at the end of the new version of Section 5 we added the Ginzburg criterion, in the $M$-layer framework, to compute the upper critical dimension. The relevance of diagrams for the two point function is discussed in Section 5 when we evaluate the possible relevant diagrams. In particular at line 359 of the new version we explain why, in this framework, additional diagrams due to 4-degree vertices do not give contribution. We also wanted to highlight the comment on the relevance of diagrams written at line 685, Appendix C, of the new version of the manuscript.

Referee: "2- Section 3 is rather cryptic for a reader who is not an expert in the field or familiar with cavity equations. Providing additional explanations would make the content more accessible to a broader audience."
Authors: As suggested we added some justification for the first two identities written in Section 3, together with an explanatory figure, Fig. 1 in the revised version. We also added a useful reference for the cavity method.

Referee: "3- On page 9, different "topological diagrams" are considered, both with three-degree vertices (G2) and four-degree vertices (G′). These diagrams turn out to be relevant, respectively, in a ϕ3 and a ϕ4 theory. At the end of the calculation, only the diagrams G1 and G2 are retained, while G′ and the tadpoles are neglected (at least for p<pc). Is this the reasoning used to argue that the upper critical dimension of the theory is Du=6, whereby the surviving diagrams are exclusively those associated with a cubic field theory?"
Authors: Diagrams $\mathcal{G}'$ and $\mathcal{G}''$ really give zero contribution for $p<p_c$, this simplification could, in principle, be used to argue that the upper critical dimension is $D_U=6$. Nevertheless we added, at the end of the new version of Section 5, the Ginzburg criterion to find $D_U$, since we think it could be useful to show it in an extended fashion.

Referee: "While the symmetry factors for the selected diagrams are immediate, it would be valuable to include a more detailed explanation of how symmetry factors are determined for more complex diagrams. I, therefore, encourage the authors to either include the calculation steps in an appendix or to refer to a suitable source. I have a similar concern about the “Bond percolation" section in which the differences/similarities are presented in an overly concise manner."
Authors: Regarding the symmetry factor, we added a reference to Appendix C of Ref. [12] in which the logic for them is explained in a designated section. In order to better explain the case of the bond percolation we decided, as also suggested by report 1, to include the bond percolation case in the second version of Section 5, in parallel with site percolation.

---

## Round 2 · Referee Report · Anonymous (Referee 1) · 2024-12-2

Report

I already gave a positive appreciation of the paper in my previous report. The authors made all the changes that I suggested. I recommend publication in the present form.

Recommendation

Publish (meets expectations and criteria for this Journal)

---

## Round 2 · Referee Report · Anonymous (Referee 2) · 2024-12-4

Report

The authors have thoroughly addressed all my previous comments.

Recommendation

Publish (meets expectations and criteria for this Journal)

---

## Round 2 · List of Changes

The authors, thanks to the suggestions of the reports, decided to make the following changes:
1) As suggested by the reports, we deleted the Section on the bond percolation case (Section 6 of the version uploaded on 17 August 2024) and we incorporated the discussion of this case in the previous Section, in order to treat site and bond percolation in parallel, since there are few differences between the two cases.
2) Thanks to report #1 we separated Section 5 of the version uploaded on 17 August 2024 (the old one) in two: the new Section 5 is about the results of the $M$-layer construction while the new Section 6 is about a standard RG method used to compute estimates of the critical exponents. Following the suggestion of report #2, at the end of the new Section 5 we also added a paragraph in order to explicitly show how to find that the upper critical dimension for percolation is $D_U=6$, following the same steps done in the same $M$-layer setting, but for a different problem (ref: Phys. Rev. Lett. 128, 075702).
3) We revised Section 3, about the percolation problem on the Bethe lattice, thanks to both the useful comments of the reports. We added few remarks on the cavity method and how it is used for this problem. We added a figure in order to explain the cavity equation written (Eq. (19) of the old version) and the related Eq. for $g(s,p)$ (Eq. (20) of the old version). Moreover, we made some minor corrections, in particular, we corrected the definition of $g(s,p)$ and we added some comments on the divergence of the two-point susceptibility on the Bethe lattice (lines 167 to 180 of the new version).

We also made the following minor corrections:
- corrected the subscript of $C$ in Eq. 5 of the old version;
- added a reference for the symmetry term at line 262 of the new version;
- added the corrections term in Eq. (11) in order to remark the perturbative nature of this relation between $\lambda$ and $u$;
- we defined the notions of "topological loop"
and "topological diagram" at lines 222 and 242 of the new version respectively;
- we corrected the statement done at the end of the conclusions Section (at line 490 of the old version): "To proceed further and obtain the values of the critical exponents in the non-percolating phase" in "To proceed further and obtain the values of the critical exponents in the percolating phase";
- in this new version we always refer to the upper critical dimension with $D_U$, we corrected at line 35 $D_c$ into $D_U$.

---

## Editorial Decision

published